# A 5 × 200 Gbps microring modulator silicon chip empowered by two-segment Z-shape junctions

Yuan Yuan ®[1] ✉, Yiwei Peng ®[1], Wayne V. Sorin ®[1], Stanley Cheung ®[1], Zhihong Huang[1], Di Liang[1], Marco Fiorentino ®[1] & Raymond G. Beausoleil[1]

Optical interconnects have been recognized as the most promising solution to accelerate data transmission in the artificial intelligence era. Benefiting from their cost-effectiveness, compact dimensions, and wavelength multiplexing capability, silicon microring resonator modulators emerge as a compelling and scalable means for optical modulation. However, the inherent trade-off between bandwidth and modulation efficiency hinders the device performance. Here we demonstrate a dense wavelength division multiplexing microring modulator array on a silicon chip with a full data rate of 1 Tb/s. By harnessing the two individual p-n junctions with an optimized Z-shape doping profile, the inherent trade-off of silicon depletion-mode modulators is greatly mitigated, allowing for higher-speed modulation with energy consumption of sub-ten fJ/bit. This state-of-the-art demonstration shows that all-silicon modulators can practically enable future 200 Gb/s/lane optical interconnects.

Increasing compute demands driven by emerging applications, such as generative artificial intelligence (AI), extended reality (XR), and full self-driving (FSD), have necessitated the need for faster and more efficient interconnect technologies. Multi-node, multi-chip systems with seamless, high-speed communication between processors are vital to supporting large-scale computing and training in this AI era[1]. Traditional intra- and inter-chip interconnects relying on electrical links are struggling to satisfy the escalating requirements for bandwidth, density, and power consumption imposed by current applications. Alternatively, optical solutions have demonstrated their superiority over conventional electrical links, leveraging the high bandwidth, length-independent impedance, and multiplexing capabilities. With exceptional bandwidth and scalability, optical interconnects emerge as a compelling solution to address the surging data transmission requirements. Silicon photonics (SiPh) technology, which advanced significantly over the past decade, has been recognized as a natural solution for chip interconnects because it shares the same material platform that enables complimentary-metal-oxide-semiconductor (CMOS) compatibility and high-volume manufacturing[2–4]. Furthermore, the accompanying co-packaged optics (CPO) approach can be applied to integrate and co-optimize electronics and photonics on a single substrate to dramatically shorten the link lengths, increase data density, and reduce energy cost[5]. The high-index contrast on the SiPh platform offers single-mode, high-confinement, low-loss waveguides that allow compact microring resonators. Harnessing the intrinsic wavelength selectivity of microring resonators, wavelength division multiplexing (WDM) can be implemented on Si chips to provide a scalable modulation solution. Consequently, Si microring resonator modulators (MRMs) provide CMOS-compatible, scalable modulation in a dense footprint, which is desired for Si chip interconnects.

Great effort has been devoted to Si MRMs[6,7]. Using the free-carrier dispersion effect, the Si MRMs can realize the electro-optic (EO) modulation via manipulation of carrier concentration. Compared to large Si Mach-Zehnder modulators (MZMs) with millimeter-scale lengths[8,9], Si MRMs boast a much smaller footprint, typically spanning tens of micrometers, because the optical resonance inside the cavity effectively increases the optical path length. The compact footprint also reduces the device's capacitance leading to higher bandwidth and lower power consumption. The interaction between carrier concentration and propagating light can be manipulated electrically, including carrier injection, accumulation, and depletion. The carrier injection mode Si MRM consists of a p-i-n junction where the

[1]Hewlett Packard Labs, Hewlett Packard Enterprise, Milpitas, CA 95035, USA. ✉e-mail: yuan.yuan@hpe.com

refractive index is changed by the forward injection of carriers[10]. While it can achieve substantial resonance shifts, the operating speed is constrained by the long lifetime of Si carriers. Despite employing complex pre-emphasized signals, the injection-mode MRM can only sustain data rates around 10 to 20 Gb/s[11]. Differently, the carrier accumulation mode MRM employs a forward-biased metal-oxide-semiconductor (MOS) capacitor. A thin insulating oxide layer (~10 nm) is sandwiched between p-type poly-Si and n-type crystalline-Si layers to form a waveguide that facilitates an overlap between the optical mode and accumulated carriers[12,13]. This structure also enables efficient modulation. The bandwidth of the accumulation mode MRM is not limited by carrier lifetime, but is instead governed by the resistance-capacitance (RC) time constant of the MOS capacitor. Additional oxide and ploy-Si depositions are also required for the carrier accumulation mode MRM. Conversely, the carrier depletion mode MRM utilizes a reverse bias voltage to exploit the plasma dispersion effect. The reverse-biased p-n junction mitigates the minority carrier lifetime limitation, resulting in faster operating speed. However, the modulation efficiency of the depletion mode MRM is comparatively smaller than the other two modes of Si MRMs. To improve the modulation efficiency, the overlap integral between the carrier density variation and the optical mode should be optimized. Distinct Si p-n junction configurations have been demonstrated for MRMs, such as the vertical junction[14-17], lateral junction[18-21], and L-shape junction[22,23]. Nevertheless, the trade-off between modulation efficiency and RC time-limited bandwidth still hinders the performance of the Si depletion mode MRM.

Here we propose a Si MRM featuring two-segment Z-shape junctions to alleviate this trade-off. The Z-shape junction profile increases the effective carrier-light interaction and reduces the device series resistance, meanwhile the two separate junctions design reduces the junction capacitance and facilitates pulse amplitude modulation with four levels (PAM4) using two non-return-to-zero (NRZ) signals. This device exhibits a high EO bandwidth of ~48.6 GHz without optical peaking effect, a good modulation efficiency $V_\pi \cdot L$ of ~0.6 V·cm, and an energy consumption of ~6.3 fJ/bit for 200 Gb/s PAM4 modulation. The 5-channel dense wavelength division multiplexing (DWDM) Si MRM array is also experimentally demonstrated that supports 1 Tb/s operation with channel crosstalk < −33 dB.

## Results

### Device design

The schematic diagram of the 5-channel DWDM Si MRM is shown in Fig. 1a, each MRM has a slightly different radius to provide a uniform distribution of resonance wavelengths within one free spectral range (FSR). The nominal radius of the MRM is 12 $\mu$m, corresponding to a $FSR \approx \lambda^2/(n_g \cdot L)$ of ~5.7 nm (1 THz). Given a DWDM channel frequency spacing of $\Delta f$, the microring circumference difference $\Delta L$ can be expressed as

$$\Delta L \approx \frac{n_g}{n_{eff}} \frac{\Delta f}{f} L, \tag{1}$$

where $n_g$ is the group refractive index, $n_{eff}$ is the effective refractive index, $f$ is the resonance frequency, and $L$ is the microring circumference. For this DWDM array, a $\Delta f$ equal to 200 GHz is chosen for 5 channels. A TiN layer situated atop the MRM is used as a heater to tune the resonance of each channel to accommodate fabrication errors. Within each MRM, two individual modulation segments, the most significant bit (MSB) and the least significant bit (LSB), are incorporated with a length ratio of ~2:1. The two-segment structure is tailored for PAM4 modulation. Traditional PAM4 modulation is realized by a PAM4 driving signal, where a power-hungry digital-to-analog converter (DAC) is essential to generate equally spaced 4 levels. However, as data rates increase, both power consumption and equalization intricacy of the PAM4 driver will increase accordingly.

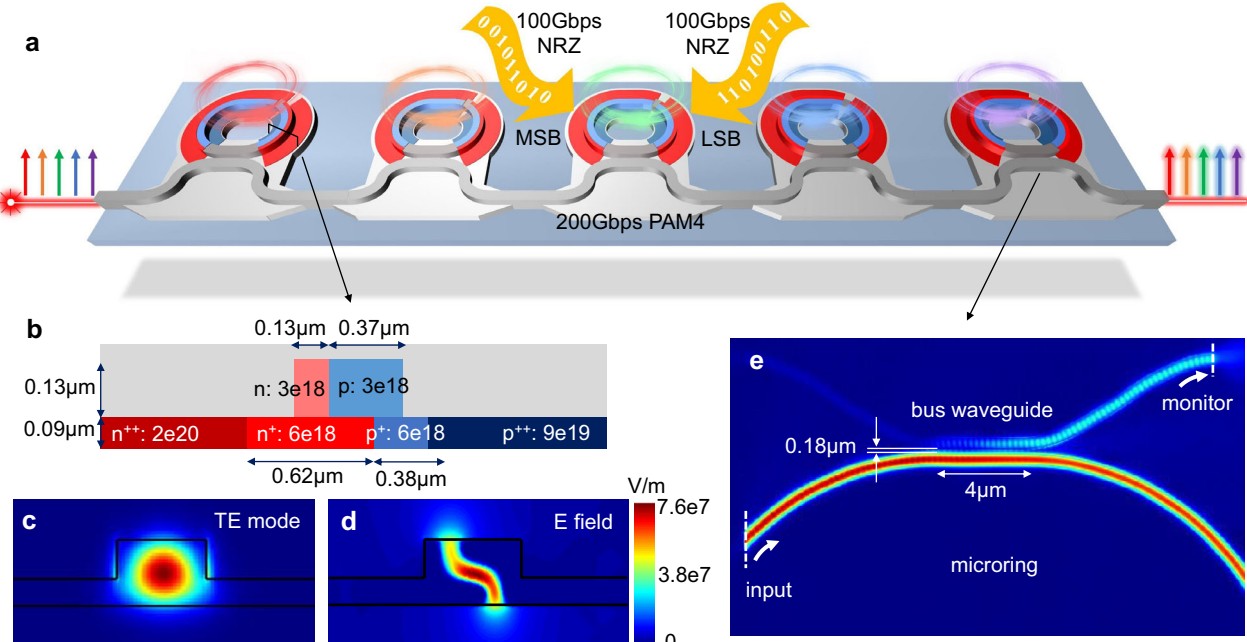

**Fig. 1 | Device design of the Si two-segment Z-shape microring modulators (MRMs). a** Schematic diagram of the 5-channel Si two-segment MRMs. **b** Cross-sectional diagram of the *Z*-shape junction with its corresponding waveguide transverse electric (TE) mode (**c**) and electric field at -3 V (**d**). **e** finite-difference time-domain (FDTD) simulated field distribution of the coupling region.

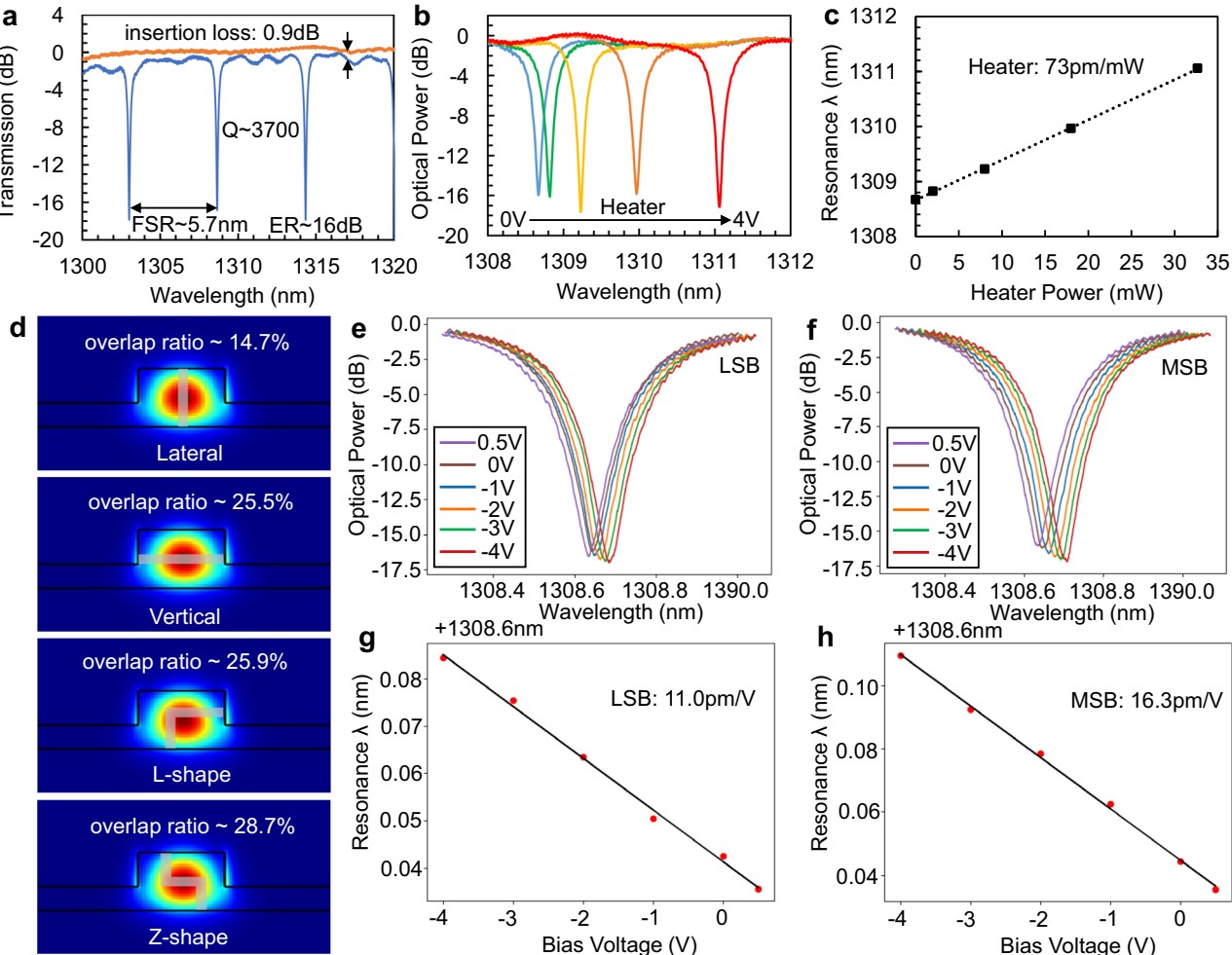

**Fig. 2 | Direct current (DC) characteristics of a single microring modulator (MRM). a** Measured transmission spectrum of a single-channel MRM (blue) and a pair of grating couplers (orange). Measured transmission spectrum versus heater voltage ranging from 0 V to 4 V **b** and resonance wavelength versus heater power. **c**, **d** Calculated overlap ratios between the waveguide transverse electric (TE) mode and depletion regions of different junctions. **e**, **f** Measured transmission spectrum versus junction voltage from 0.5 V to -4 V at the least significant bit (LSB) and most significant bit (MSB). **g**, **h** Extracted resonance wavelengths versus junction voltage for LSB and MSB.

PAM4 for 200 Gb/s per lane is proposed in IEEE 802.3 Standards for the next generation 800 Gb/s and 1.6 Tb/s Ethernet. Generating 200 Gb/s PAM4 signals with equally spaced levels on an electronic chip is exceedingly challenging. To address this issue, the two-segment structure functions as an optical DAC. The 2:1 length ratio of the two p-n junctions enables two pairs of modulation levels [0, 1] and [0, 2], thus the PAM4 levels [0, 1, 2, 3] can be easily achieved in the optical domain with two electrical NRZ signals. The linearity of the optical DAC structure is detailed in Supplementary Information IV.

The cross section of the p-n junctions is shown in Fig. 1b, which is based on a standard 500 nm-wide, 220 nm-thick Si waveguide with a 90 nm-thick slab. A $Z$-shape doping profile is devised for the p-n junctions incorporating 4 different implantations: $n$-type doping with the concentration of $3 \times 10^{18}$ cm$^{-3}$, p-type doping of $3 \times 10^{18}$ cm$^{-3}$, $n^+$-type doping of $6 \times 10^{18}$ cm$^{-3}$, and $p^+$-type doping of $6 \times 10^{18}$ cm$^{-3}$. By tuning the implant ion energy, the n- and p-type dopings are located in the upper 130 nm region, while the $n^+$- and $p^+$-type dopings are positioned in the lower 90 nm section. These dopings result in a small series resistance of the slab region while maintaining a favorable balance between modulation efficiency and free carrier absorption within the waveguide core region. Furthermore, the 4 dopings exhibit a horizontal width offset. This offset enables a more effective modulation by capitalizing on the fact that holes in Si possess a greater real-part index change and lower absorption loss compared to electrons. The simulated optical transverse electric (TE) mode and electric field distribution at -3 V within this Si waveguide are illustrated in Fig. 1c, d, respectively. It is clear that the highest intensity region of the TE mode aligns well with the depletion region of the $Z$-shape junction, thereby optimizing the modulation efficiency. The coupling region of the Si MRM is also designed based on the loss simulation of the two-segment $Z$-shape junctions. We choose a 4 $\mu$m-long coupling region with a gap of 180 nm. The 3D finite-difference time-domain (FDTD) simulation of the coupler is depicted in Fig. 1e.

**Single microring modulator: direct current characterizations**

To evaluate and analyze the performance of this design, we first performed measurements on a single-channel MRM. Figure 2a shows the measured transmission spectrum of a single-channel MRM (represented by the blue curve) and a single pair of grating couplers (represented by the orange curve). The comparison of two curves reveals that the average off-resonance insertion loss of the MRM is ~0.9 dB. Besides, the MRM spectrum indicates that the device exhibits an FSR of ~5.7 nm, consistent with the design value, a direct current (DC) extinction ratio (ER) of ~16 dB, a full width at half-maximum (FWHM) of ~0.35 nm, and a quality factor (Q) of ~3700. The finesse of the MRM thus can be extracted as $F = FSR/FWHM \approx 16$, which is sufficient to support 5 channels for DWDM. The MRM

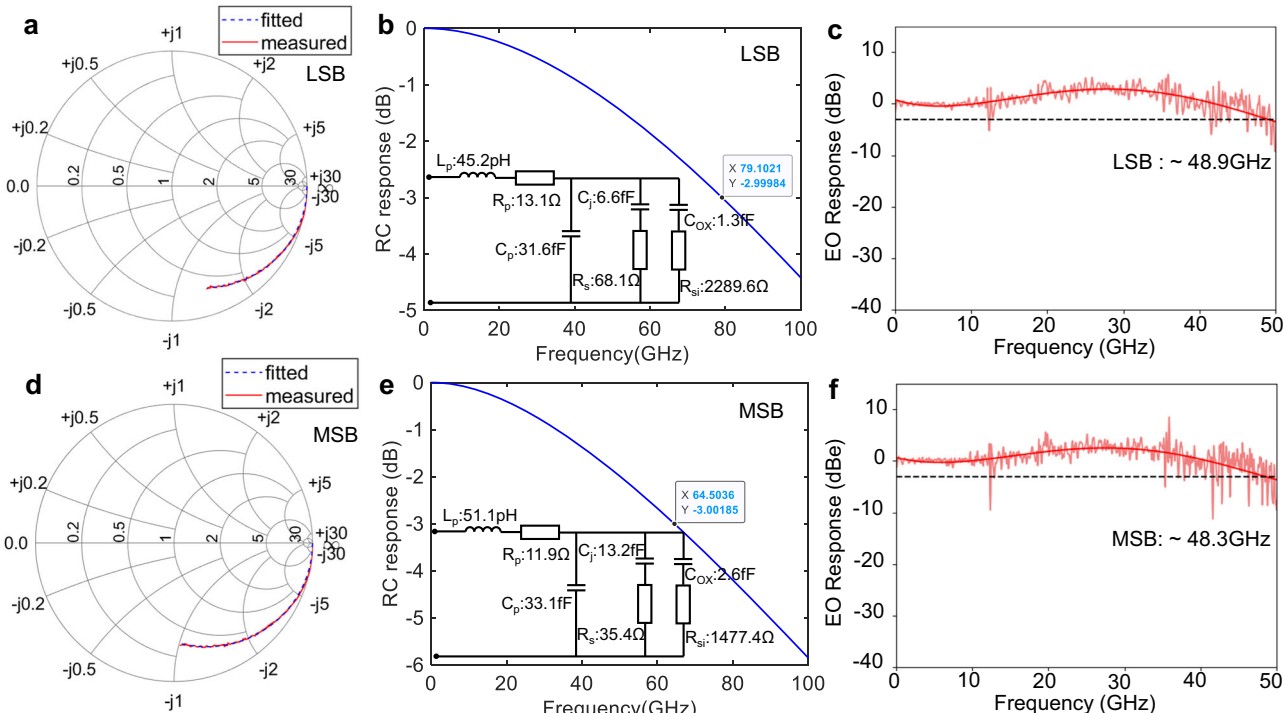

**Fig. 3 | Frequency responses of a two-segment Si microring modulator (MRM). a, d** Measured (red) and fitted (blue) Smith charts of S11 for the least significant bit (LSB) and most significant bit (MSB) at -3 V. **b, e,** Fitted equivalent circuits and corresponding junction resistance-capacitance (RC) responses for LSB and MSB at -3 V. **c, f** Measured electro-optic (EO) responses (|S21|²) for LSB and MSB with detuning of Δλ - 0.1 nm at −3 V.

waveguide loss is analyzed in Supplementary Information I. The MRM transmission spectrum versus the TiN heater bias voltage, spanning from 0 V to 4 V, is plotted in Fig. 2b. The integrated TiN heater has a resistance of -500 Ω. The resonance red-shifts as voltage raises, and the corresponding resonance wavelength against heater power is shown in Fig. 2c, where the red-shift is -73 pm/mW. For a $\pi$ phase shift about 39 mW heater power is required.

As described in the Device design section, a tailored Z-shape junction has been designed to optimize modulation efficiency. The EO overlap ratios of diverse p-n junctions are illustrated in Fig. 2d, arranged from top to bottom as lateral junction, vertical junction, L-shape junction, and Z-shape junction. Assuming these p-n junctions have an equivalent depletion region width of 40 nm, then the integrals of the optical TE mode within the shaded depletion regions will be -14.7%, 25.5%, 25.9%, and 28.7%, respectively. The Z-shape junction allows the highest overlap ratio. The Z-shape junction requires only conventional ion implantations, which are fully compatible with the standard fabrication process of SiPh foundries. Additionally, the 4 implantations provide more freedom to simultaneously optimize the modulation efficiency, free carrier absorption loss, and series resistance. The modulation efficiency of the Z-shape junction is characterized using the transmission spectrum as a function of the junction bias. Figure 2e, f show the transmission spectra of the Z-shape MRM with bias voltage ranging from 0.5 V to -4 V at the LSB and MSB junctions, respectively. The resonance wavelength shifts to the red side with higher reverse voltage. The corresponding resonance wavelengths versus bias voltage are plotted in Fig. 2g, h, showcasing a shift Δλ/ΔV of -11.0 pm/V for the LSB segment and -16.3 pm/V for the MSB segment. The figure of merit for modulation efficiency, $V_\pi \cdot L$, then can be calculated by

$$V_\pi \cdot L = \frac{FSR \cdot L_{seg}}{2\Delta\lambda/\Delta V},\tag{2}$$

where $L_{seg}$ is the segment length. Therefore, the $V_\pi \cdot L$ is -0.53 V·cm for the LSB segment and -0.69 V·cm for the MSB segment. Compared to the lateral junction two-segment MRM with a $V_\pi \cdot L$ of 1.0 V·cm[21], the Z-shape junction improves the modulation efficiency by -67%.

**Single microring modulator: radio frequency characterizations**
Alongside the modulation efficiency, the modulation bandwidth is another critical metric of the MRM, dictating the achievable data rate. The EO bandwidth of the MRM is determined by the RC time constant of the junction, the photon lifetime of the microring, and the wavelength detuning from the resonance. The RC time constant can be extracted from the S11 results. A 50 GHz vector network analyzer was used for the S-parameter measurements, and all other components were calibrated. The Smith charts in Fig. 3a and d visualize the measured S11 for the LSB and MSB at −3 V, represented by the red solid lines. The equivalent circuits for both segments then can be derived through the S11 fitting. The fitted S11 is represented by the blue dash lines in the Smith charts, exhibiting good agreement with the measured data. The fitted equivalent circuits for both segments are shown as insets in Fig. 3b and e. In these circuits, $C_j$ denotes the p-n junction capacitance, $R_s$ is the p-n junction series resistance, $C_{OX}$ is the buried oxide capacitance, $R_{si}$ stands for the Si substrate resistance, and $L_p, R_p, C_p$ encompass the inductance, resistance, and capacitance of the parasitic parameters from vias and pads. For the LSB, the Z-shape p-n junction has a capacitance $C_j$ of -6.6 fF and a resistance $R_s$ of -68.1 Ω. For the MSB, the corresponding values are $C_j$ - 13.2 fF and $R_s$ - 35.4 Ω. As expected, the $C_j$ of the MSB is twice that of the LSB, while the $R_s$ of the MSB is half that of the LSB due to the segment length ratio of 2:1. The $C_j$ and $R_s$ values are very close to the theoretically calculated results based on the doping profile. Utilizing the equivalent circuits with a 50 Ω source impedance, the frequency response across the p-n junction capacitance can be plotted, which corresponds to the signal response that is actually effective for the modulation. As shown in Fig. 3b, the RC time-limited bandwidth of the LSB is -79.1 GHz. On the other hand, the MSB has an RC bandwidth $f_{RC}$ of -64. GHz, illustrated in

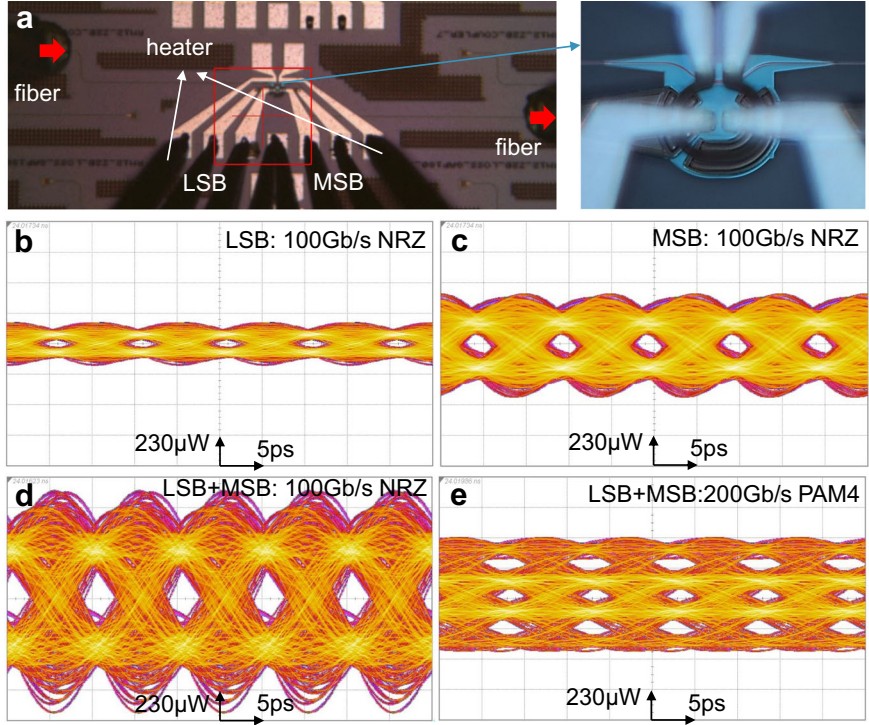

**Fig. 4 | Eye diagrams of a single microring modulator (MRM). a** Micrograph of a two-segment Si MRM. Measured eye diagrams of 100 Gb/s non-return-to-zero (NRZ) of the least significant bit (LSB), most significant bit (MSB), both LSB and MSB (**b**–**d**); and 200 Gb/s pulse amplitude modulation with four levels (PAM4) of both LSB and MSB (**e**).

Fig. 3e. Despite the nearly constant product of $R_s \times C_j$ for different segment lengths, the shorter junction exhibits higher RC bandwidth considering the entire equivalent circuits. For comparison, the one-segment MRM with an assumed $C_j$ of 19.8 fF and $R_s$ of 23.3 Ω would yield an RC bandwidth of ~53.6 GHz. The two-segment design not only simplifies the CMOS driver but also improves the junction RC-limited bandwidth by >20%.

The second EO bandwidth contributor, the photon lifetime of the microring $\tau_{ph}$, is characterized by the Q factor of the resonant cavity. At critical coupling, the photons need to travel $1/(2\alpha)$ distance to decay to $1/e$, therefore $\tau_{ph} = n_g/(2c\alpha)$[24]. The photon lifetime-limited bandwidth, $f_{ph}$, then can be expressed by

$$f_{ph} = \frac{1}{2\pi\tau_{ph}} = \frac{c\alpha}{n_g\pi} \approx \frac{c}{\lambda Q}, \qquad (3)$$

where $c$ is the light speed, $\alpha$ is the optical power loss coefficient, and $\lambda$ is the resonance wavelength. The Q factor of the MRM is 3700, so that the $f_{ph}$ is around 62 GHz. The third parameter influencing the EO bandwidth is the wavelength detuning. The MRM can achieve an extended 3 dB bandwidth with a wavelength further away from the resonance[25,26]. However, the MRM transmission spectrum follows a Lorentzian shape, whose slope is a nonlinear function of wavelength. The 3 dB bandwidth enhancement from optical peaking inevitably suppresses the response amplitude at low frequencies. Because of this reduced modulation slope, the optical peaking effect will not help the optical modulation amplitude (OMA) and ER in eye diagrams. The maximum modulation slope of the MRM occurs at the wavelength detuning of $0.29 \times$ FWHM. Given the resonance width FWHM of 0.35 nm for this MRM, the wavelength detuning we choose for the following measurements is set at $\Delta\lambda$ of 0.1 nm (-17.5 GHz), which is located at the -6 dB insertion loss (IL) point on the transmission spectrum. Figure 3c and f display the measured EO responses ($|S21|^2$) for the LSB and MSB segments, respectively. The LSB junction has a 3

dB EO bandwidth of ~48.9 GHz, and the EO bandwidth is ~48.3 GHz for the MSB junction. These two EO bandwidth values align with the estimated bandwidth from the RC-limited bandwidth and photon lifetime-limited bandwidth, $f_{est} = f_{RC}f_{ph}/\sqrt{f_{RC}^2 + f_{ph}^2}$. Using the optical peaking effect, the EO bandwidth of the MRM can be further increased. For instance, this MRM exhibits a 3 dB bandwidth of ~58 GHz at −3 dB IL, as shown in Supplementary Information II.

Thanks to the high modulation efficiency and high EO bandwidth empowered by the combination of the two-segment and Z-shape structures, this Si MRM has the potential for 200 Gb/s PAM4 modulation. A micrograph of the single-channel Si MRM is shown in Fig. 4a, a continuous wave (CW) laser is used as the input source, and the light is modulated by the LSB and MSB with two NRZ signals. The detailed experimental measurements of the eye diagrams are described in the Methods section. The measured eye diagrams at 100 Gb/s NRZ with pseudorandom bit sequence 9 (PRBS9) signals are shown in Fig. 4b, c for the LSB- and MSB-only modulations, respectively. The LSB eye diagrams exhibit a signal-to-noise ratio (SNR) of ~4.02. Due to the twice segment length of the MSB, its eye diagrams show almost double OMA and the SNR is ~4.25. The combined eye diagrams of the entire MRM can be switched between NRZ and PAM4 by adjusting the relative time offset of the two NRZ driving signals. When there is no relative time difference, the two-segment MRM allows larger NRZ eye diagrams. As shown in Fig. 4d, the combined 100 Gb/s NRZ eyes have the largest OMA and the SNR is ~4.53. In contrast, the output of the MRM will be PAM4 eye diagrams with an integer-bit time offset. Figure 4e is the 200 Gb/s PAM4 eye diagrams with a 2-bit relative time offset. The transmitter dispersion eye closure quaternary (TDECQ) is measured with a soft-decision forward error correction (SD-FEC) threshold of symbol error rate (SER) at 1E-2[23,27,28]. At the receiver side, a half baud rate 4th-order Bessel filter and a 21-tap feed-forward equalizer (FFE) were used to reduce noise. The measured TDECQ of the 200 Gb/s PAM4 eye diagrams is 0.2 dB. The outer ER, which denotes the ER

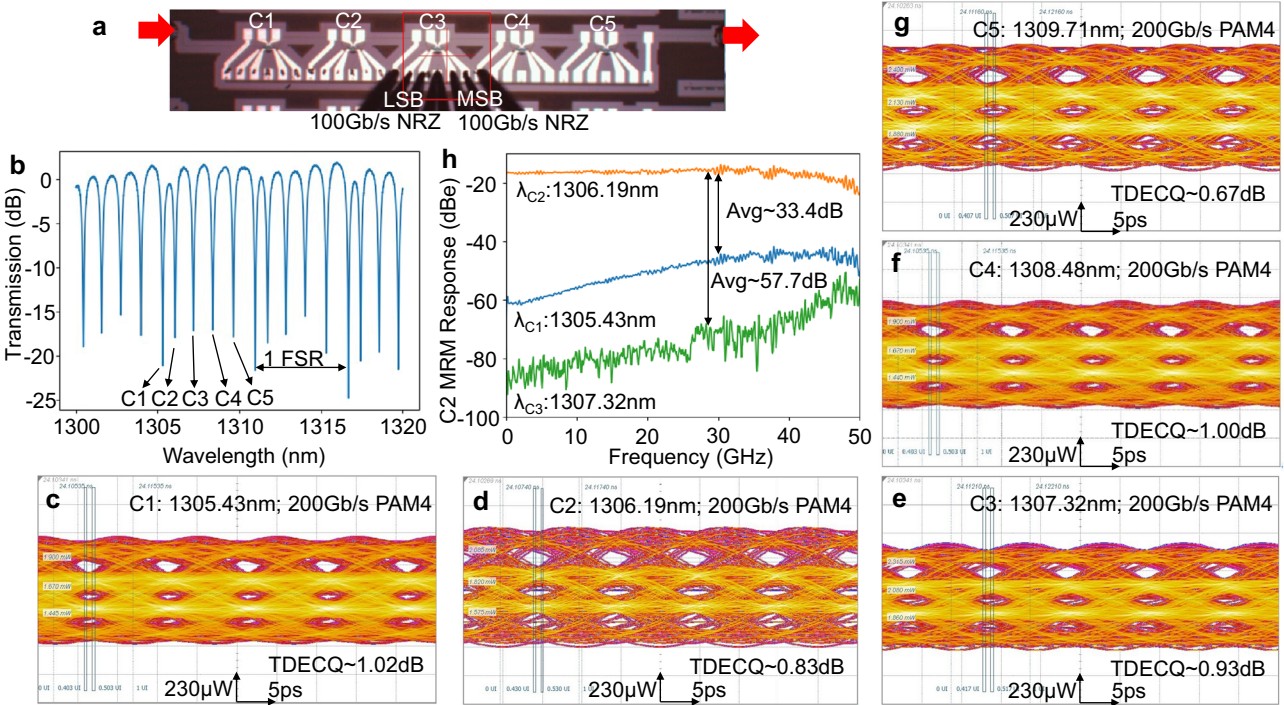

**Fig. 5 | Eye diagrams and crosstalk performances of the 5-channel Si microring modulator (MRM) array. a** Micrograph of a 5-channel dense wavelength division multiplexing (DWDM) Si MRM array. **b** Measured transmission spectrum of the 5-channel Si MRM array. **c–g** Measured eye diagrams of 200 Gb/s pulse amplitude modulation with four levels (PAM4) of channel 1, channel 2, channel 3, channel 4, and channel 5 (C1-C5). **h** Measured channel crosstalk of the DWDM MRM array.

between the top level and the bottom level of the PAM4 eye diagrams, is measured to be 3.6 dB. All these eye diagrams are based on 100 Gb/s NRZ signals with a peak-to-peak radio frequency (RF) swing voltage $V_{pp}$ of 1.6 V. The power consumption per bit $E_{bit}$ of the two-segment MRM can be calculated by

$$E_{bit} = \frac{C_{j,LSB} \cdot V_{pp}^2 + C_{j,MSB} \cdot V_{pp}^2}{2^2 \cdot \log_2(N)}, \qquad (4)$$

where $C_{j,LSB}$ and $C_{j,MSB}$ are the junction capacitance for the LSB and MSB, $2^2$ is the number of possible transitions, and $\log_2(N)$ is the bit number per symbol[29]. For PAM4 modulation, $N = 4$. Given the $C_{j,LSB} \approx 6.6$ fF, $C_{j,MSB} \approx 13.2$ fF, and $V_{pp} = 1.6$ V, the power consumption of the two-segment Si MRM is ~6.3 fJ/bit.

**Dense wavelength division multiplexing microring modulators**
The single-channel Si MRM can be seamlessly expanded into a 5-channel MRM array by slightly modifying the microring circumference according to Eq. (1). A micrograph of a 5-channel DWDM MRM array is shown in Fig. 5a, where 5 microrings are connected in series and share a common bus waveguide. The measured transmission spectrum of the 5-channel MRM array at zero bias is depicted in Fig. 5b, channel 1 through channel 5 are approximately evenly distributed within one FSR without heater tuning. Thanks to this uniform resonance distribution, each channel MRM can be directly modulated without thermal shifting of the resonance wavelength. Using the same experimental setup as the single-channel MRM eye diagrams, the 200 Gb/s PAM4 eye diagrams are measured for all 5 channels, as shown in Fig. 5c–g. The measured wavelengths of all channels were blue-shifted by 0.1 nm from their respective resonance wavelengths to optimize the OMA. The channel 1 MRM was measured at 1305.43 nm, and the measured TDECQ (at SER = 1E-2) with the Bessel filter and 21-tap FFE is ~1.02 dB. These values for the other channels are as follows: $\lambda_{C2} = 1306.19$ nm, TDECQ$_{C2}$ ~ 0.83 dB; $\lambda_{C3} = 1307.32$ nm, TDECQ$_{C3}$ ~ 0.93 dB; $\lambda_{C4} = 1308.48$ nm, TDECQ$_{C4}$ ~ 1.00 dB; and $\lambda_{C5} = 1309.71$ nm, TDECQ$_{C5}$ ~ 0.67 dB.

The measured outer ERs for all channels are -3.6 dB. All 5 channels exhibit very similar 200 Gb/s PAM4 eye diagrams with clear openings and favorable TDECQ values, the entire DWDM MRM array thus adds up to support 1 Tb/s data rates. It is worth noting that 200 Gb/s eyes are the upper limit of the experimental setup, higher bandwidth setup can further improve the data rate.

The channel crosstalk is another pivotal figure of merit for the DWDM modulators. In order to quantify the optical crosstalk, the EO response ($|S21|^2$) of an MRM was measured at adjacent and its own channel wavelengths. Channel 2 MRM is chosen here because it has the narrowest channel spacing from its neighboring channels. The measured channel crosstalk is illustrated in Fig. 5h, the orange line is the measured EO response of the channel 2 MRM at its own wavelength $\lambda_{C2}$, displaying the anticipated flat, strong response and high 3 dB bandwidth; the blue line represents the channel 2 MRM response at $\lambda_{C1}$; and the green line is the response of the channel 2 MRM at $\lambda_{C3}$. The wavelength difference between channels 1 and 2 is 1306.19 − 1305.43 = 0.76 nm, which is slightly less than the ideal channel spacing of 1.14 nm (~200 GHz) due to the fabrication variance. This narrower spacing is also evident in the transmission spectrum in Fig. 5b. Despite the 0.76 nm channel spacing, on average, the channel 1 wavelength exhibits ~−33.4 dB smaller EO response than that of channel 2. On the other hand, the wavelength difference between channels 2 and 3 is 1307.32 − 1306.19 = 1.13 nm, aligning closely with the ideal channel spacing. Consequently, the channel 3 wavelength has an even weaker EO response, which is ~−57.7 dB less than the actual channel response. The increased EO responses at high frequencies for channel 1 and 3 wavelengths are due to the optical peaking effect from the wavelength detuning. In short, this DWDM Si MRM array demonstrates remarkably small crosstalk, even without any thermal tuning, it exhibits <−33 dB crosstalk between the two closest channels. This value can be further improved to ~−57 dB level by tuning the channel spacing to the ideal 1.14 nm. Furthermore, the characterization of large-signal crosstalk was conducted by simultaneously operating channel 2 MSB and channel 3 LSB, as illustrated in Supplementary

**Table 1 | Properties of depletion mode MRMs based on different types of Si p-n junctions**

| Ref. | Junction type | Modulation structure | Radius (µm) | Q | Bandwidth (GHz) | $V_\pi \cdot L$ (V·cm) | Operating λ (nm) | Data rate (Gb/s) | Energy cost (fJ/bit) |
|---|---|---|---|---|---|---|---|---|---|
| 14 | vertical | one-segment | 4.8 | 6800 | 21 | - | 1550 | 1×44 (NRZ) | 0.9 |
| 15 | vertical | one-segment | 5 | 6780 | 24 | - | 1310 | 1×40 (NRZ) | 4 |
| 16 | vertical | one-segment | 5 | 4800 | 35 | 0.37† | 1310 | 1×100 (PAM4) | - |
| 17 | vertical | one-segment | 8 | 5600 | 67 (-3dB IL) | 0.8 | 1310 | 1×200 (PAM4) | - |
| 18 | lateral | one-segment | 3 | 1050 | 67 | 0.94 | 1550 | 1×200 (PAM4) | - |
| 19 | lateral | one-segment | 8 | 4500 | 52 | 0.825 | 1310 | 1×240 (PAM8) | - |
| 20 | lateral | two-segment | 12 | 7500 | 18 | - | 1310 | 1×40 (PAM4) | - |
| 21 | lateral | two-segment | 12 | 3600 | 40 | 1.0 | 1310 | 1×100 (PAM4) | 9.9 |
| 22 | L-shape | one-segment | 10 | 5000 | 50 | 0.52 | 1310 | 1×128 (PAM4) | 18 |
| 23 | L-shape | one-segment | 4 | 4000 | 54 | 0.53 | 1310 | 1×240 (PAM4) | - |
| 30 | U-shape | one-segment | 32.5 | 26200 | 13.5 | 0.46 | 1310 | 1×13 (NRZ) | - |
| 31 | Z-shape | one-segment | 9 | 7000 | 25 (-6dB IL) 42 (-3dB IL) | 0.92 | 1310 | 1×100 (PAM4) | 14 |
| this work | Z-shape | two-segment | 12 | 3700 | 48.6 (-6dB IL) 58.1 (-3dB IL) | 0.6 | 1310 | 5×200* (PAM4) | 6.3 |

† 0.37 V·cm is measured from 0.5 V to -0.5 V; $V_\pi \cdot L$ would increase to > 0.53 V·cm in the higher reverse voltage region.

* 200 Gb/s limited by the bandwidth of the experimental setup.

Information IV. At a data rate of 100 Gbaud/s, the eye quality remains unaffected, affirming the negligible impact of channel crosstalk.

## Discussion

In this work, a Si MRM with two-segment Z-shape junctions has been demonstrated for use in the next-generation optical interconnects. The properties comparison of depletion mode MRMs with diverse Si p-n junctions, including vertical, lateral, L-shape, U-shape, and Z-shape junctions, are summarized in Table 1. To enhance modulation efficiency, longer depletion regions have been adopted for better mode overlap, such as the vertical, L-shape, and U-shape p-n junctions. The U-shape junction offers the best modulation efficiency $V_\pi \cdot L$ of 0.46 V·cm. However, it also results in the longest junction capacitor, nearly twice as long as the L-shape and Z-shape junctions. The ensuing large RC time constant will hinder the high-speed operation. One potential solution to mitigate this RC issue is to lower the doping concentrations in the U-shape junction, allowing the middle p-type layer to be almost entirely depleted, thereby significantly reducing the junction capacitance[30]. Unfortunately, the low doping concentrations result in a much higher Q factor, which in turn restricts the photon lifetime-limited bandwidth. Consequently, the U-shape design is more suitable for low-speed, high-efficiency modulation. Other than that, L-shape junctions exhibit the second-best modulation efficiency[22]. The simulated Z-shape junction is expected to offer ~11% better overlap compared to L-shape junctions, but the experimental results do not fully reflect this. This discrepancy can be attributed to the fact that the actual fabricated Z-shape doping profile deviates from the ideal design. As the first batch of Z-shape MRMs, its modulation efficiency is already close to that of other state-of-the-art MRMs. Further optimization of the Z-shape doping profile holds the potential for achieving higher modulation efficiency $V_\pi \cdot L$ < 0.5 V·cm. Besides the enhanced overlap between the waveguide mode and the junction depletion region, our innovative design offers high bandwidth at the same time. The 4 distinct implantations create higher doping concentrations inside the slab region than the waveguide core region to achieve a reduced series resistance without leading to high free carrier absorption loss. With this unique degree of freedom, our Z-shape MRM exhibits a similarly high bandwidth even with larger microring radius. Meanwhile, the two-segment structure is employed to further reduce the capacitance of the Z-shape junctions. Therefore, this MRM combines a high RC-limited bandwidth with better modulation efficiency. In addition, the two-segment architecture simplifies the complex PAM4 driving signals into two NRZ signals. With the adoption of this optical DAC, the MRM can modulate higher-speed data with better linearity and lower CMOS driver energy consumption.

Thanks to the innovative design, our two-segment Z-shape MRM exhibits a high 3dB bandwidth of ~48.6 GHz at the wavelength with maximum modulation slope, a large modulation efficiency with $V_\pi \cdot L$ of ~0.6 V·cm, and thus supports open 200 Gb/s PAM4 eye diagrams with only 1.6 V $V_{pp}$ and 6.3 fJ/bit energy consumption. The demonstrated MRM achieves state-of-the-art performance for Si depletion mode modulators. Compared to the previous leading two-segment MRM with a lateral junction[21], it enables a ~21% increase in bandwidth, a remarkable ~67% enhancement in modulation efficiency, a doubling of the data rate, and a ~36% reduction in energy cost per bit. With this high performance, the 5-channel DWDM Si MRM array based on two-segment Z-shape junctions has been realized, which supports 1 Tb/s (5 × 200 Gb/s PAM4) data rates. The inherent offset resonances of the microrings allow the DWDM MRM array to act as an effective multiplexer (MUX), achieving an impressively low crosstalk of <-33 dB directly, with the potential for further improvement to ~−57 dB with control circuits. The DWDM structure provides scalability for ever-increasing data rates. By reducing the MRM radius to 5 µm, the FSR of the MRM can be extended to ~13.7 nm with a bend loss of only about 0.06 dB. With the same channel spacing, 12 channels can be supported, easily scaling up the full data rate to 2.4 THz. To encapsulate, this work showcases a total of 1 Tb/s DWDM modulation integrated on a Si chip with a small $V_{pp}$ of 1.6 V, paving the way to satisfy the ever-increasing data demands for intar- and inter-chip interconnects.

## Methods

### Fabrication

The Si MRM chips were fabricated at Advanced Micro Foundry (AMF), Singapore. The chips are based on industry-standard 220 nm thick Silicon-on-Insulator (SOI) wafers. The Z-shape junction was formed from Phosphorous (for n-type) and Boron (for p-type) implants. The uniformity of the Si MRM chips is shown in Supplementary Information V.

### S-parameter measurements

All S-parameter characteristics were measured with the 50 GHz vector network analyzer (VNA) N5225A. In S11 measurements, an impedance standard substrate was used to de-embed the RF probe, connectors, cables, and bias-tee. In S21 measurements, the RF connectors, cables,

and bias-tee were calibrated by the electronic calibration module N4693-6003. The S21 response of the MRM was then captured by the 40 GHz photodiode RXM40AF. The frequency response of the photodiode was obtained from its measured impulse response, and it was used to compensate for the measured S21 data to achieve the frequency response of the MRM alone.

## Eye diagrams measurements

The eye diagrams of the MRMs have been performed with a 120 GSa/s arbitrary waveform generator (AWG) M8194A, which can generate frequency content up to 50 GHz. It provides two PRBS9 NRZ signals for the LSB and MSB. A signal generator was used to generate a 2.5 GHz clock signal for the AWG clock reference, the digital communication analyzer (DCA) trigger reference, and the DCA precision reference. The highest swing voltage $V_{pp}$ of the AWG is 0.8 V, therefore two identical 60 GHz electrical power amplifiers were added to amplify the swing voltage. After that, two identical 50 GHz bias-tees were connected to combine the RF signals and DC bias voltage. The measured $V_{pp}$ after the bias-tee at 100 Gb/s NRZ, i.e. the LSB and MSB driving voltage, is 1.6 V. A commercial tunable CW laser, Santec TSL 510, was used as the input optical source, and the MRM output light was amplified by a Praseodymium-doped fiber amplifier (PDFA) to compensate for the link losses. A 65 GHz optical module N1030A was used to receive the modulated signals on the DCA. The measurement setup is illustrated in Supplementary Information VI. The signal distortion caused by the power amplifier, bias-tee, and RF cables was calibrated by the AWG internal calibration tool. In order to eliminate the amplified spontaneous emission (ASE) noise from the PDFA, the received patterns were averaged 64 times. A more than half-baud rate 4th-order Bessel filter and a 21-tap FFE were used at the receiver. The TDECQ values of the 200 Gb/s PAM4 eye diagrams were measured at an SD-FEC threshold of SER at 1E-2. The 200 Gb/s PAM4 is the upper limit of the current eye diagram experimental setup, a higher data rate is achievable with higher bandwidth AWG and RF components.

## Data availability

The data that support the findings of this study are available from the corresponding author upon request.

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

## Acknowledgements

We thank Mrs. Xuema Li for taking the scanning electron microscope (SEM) pictures.

## Author contributions

Y.Y. designed the devices, taped out the chips, performed the measurements, and wrote the manuscript. Y.P. participated in the chip tape-out and measurements, W.V.S. and S.C. participated in the measurements, Z.H., D.L., M.F. and R.G.B. supervised the study and gave important technical advice. All authors reviewed the manuscript.

## Competing interests

The authors declare no competing interests.
