## [Peer Review File · Nature Communications]

A 5×200 Gbps DWDM Microring Modulator Silicon Chip Empowered by Two-Segment Z-Shape JunctionsREVIEWER COMMENTS

Reviewer #1 (Remarks to the Author):

This paper has demonstrated a DWDM microring modulator array on silicon with a total data rate of 1 Tb/s. Specially, it harnesses two individual p-n junctions with an optimized Z-shape doping profile and achieves a high bandwidth of ~ 48.6 GHz as well as a high modulation efficiency with $V_n \cdot L$ of ~ 0.6 V·cm. Open eye diagrams at 200 Gb/s PAM4 have been achieved with a swing voltage of 1.6 V and energy consumption of 6.3 fJ/bit. This work is pretty nice and the following comments should be addressed before it is to be accepted.

1. Could the authors show the measurement results for more samples of the same batch, which is important to show the uniformity of the fabrication.
2. As the authors mentioned, the spacing between channels 1 and 2 is 0.76 nm, which is actually 66.6% of the desired channel spacing of 1.14 nm. Therefore, the claim of "without heater tuning, channel 1 through channel 5 are already uniformly distributed within one FSR" is convinced and should be modified.
3. The bending radius is chosen as 12 μm and the corresponding FSR is about 5.7 nm, which is quite small if more channels are involved. How much is the bending loss for the present design? Could the radius be reduced? Please give a comment on this.
4. Enlarged views for the MRM and the MRM array should be provided in Fig. 4(a) and Fig. 5(a).
5. What kind of laser source used in the experiment with the MRM array? Is it a multi-wavelength laser?
6. The authors claimed "this DWDM Si MRM array demonstrates remarkably small crosstalk, even without any thermal tuning, it exhibits < -33 dB crosstalk between the two closest channels. This value can be further improved to ~ -57 dB level by tuning the channel spacing to the ideal 1.14 nm." Please give more details to explain the crosstalk (Fig. 5h) from which to which. Is it optical or electrical crosstalk?
7. How about the real loss of the MRR waveguide in experiment?
8. The eye-diagram quality seems not very high. Please give an explanation and a discussion on how to improve it.

Reviewer #2 (Remarks to the Author):

In this manuscript, the authors demonstrated a 1 Tbps DWDM microring modulator silicon chip empowered by two-segment z-shape junctions. Each silicon depletion-mode modulator achieved a bandwidth of ~ 48.6 GHz and a modulation efficiency with $V_n \cdot L$ of ~ 0.6 V·cm. Although a two-segment Z-Shape junction is introduced, and a dense wavelength division multiplexing microring modulator array is demonstrated, I do not recommend this manuscript to be accepted in Nature Communications under its current version due to the following reasons.

1. In general, the innovation of this article is insufficient. The optical DAC, Z-shape, and DWDM microring modulator array have been reported in other papers, and the performance of a single depletion-mode microring modulator is close to or even slightly better than that of this paper, as shown in Table 1.
2. The article has mentioned many times that "the two segment structure is employed to reduce the capacitance of the Z-shape junctions", what is the benefit of reducing the junction capacitance? The authors seem to want to express that the bandwidth can be improved, but from Figure 3, the junction capacitance of the MSB is larger than that of the LSB, yet the measured bandwidth in both segments is almost the same.
3. The optical DAC needs to pay attention to its linearity, but the electrical crosstalk and the linearity are not characterized in this paper.
4. In the manuscript, the SER used in the measurement of TDECQ is $1e-2$. What is the basis for this? The current used in the industry is KP4 encoding, which has a maximum error correction capability of $2e-4$.

Reviewer #3 (Remarks to the Author):

This paper reports a silicon micro-ring modulator (MRM)-based optical transmitter chip capable of supporting 1 Tbps data transmission. The silicon MRM features a Z-shaped p-n junction with two segments designed for PAM4 modulation at 200 Gbps. The paper demonstrates an array of 5 such MRMs in series, forming a WDM transmitter with a projected capacity of 1 Tbps for data transmission.

The paper is well-written, and it provides a detailed description of the MRM and its characterization. However, there are several concerns that need to be addressed for this paper to meet the expected quality standards of Nature Communications.

The Z-shaped p-n junction is anticipated to offer greater overlap between the optical mode and the depletion region, potentially resulting in higher modulation phase efficiency when compared to other junction shapes, as shown in Fig. 2. However, the experimental measurements do not align with the simulation results. Specifically, the measured phase efficiency of 0.6 V·cm falls short of that achieved by the vertical junction at 0.37 V·cm (ref. 16), the L-shaped junction at 0.52 V·cm (ref. 22), or the U-shaped junction at 0.46 V·cm (<https://doi.org/10.1364/OE.25.008425>), which was not cited. A comprehensive discussion is essential to help readers comprehend the disparities between experimental results and theoretical predictions, as well as the technical advancements compared to prior research.

The electro-optic (EO) bandwidth and the data rate achieved with the MRM do not rank as the best when compared to some of the previously published results (as shown in table 1). A strong justification is needed for this paper to be published in Nature Communications.

The claimed data rate of 1 Tbps is a projected capability of the MRM array rather than an experimentally demonstrated data transmission with all 5 MRMs in operation. Additionally, the crosstalk measurements do not reflect real operational conditions, where each MRM would be carrying 200 Gbps of data.

The authors should review and consider these points to improve the quality and credibility of the paper. This will also help in justifying its publication in Nature Communications over other technical journals."

Reviewer #1

This paper has demonstrated a DWDM microring modulator array on silicon with a total data rate of 1 Tb/s. Specially, it harnesses two individual p-n junctions with an optimized Z-shape doping profile and achieves a high bandwidth of ~ 48.6 GHz as well as a high modulation efficiency with $V\pi \cdot L$ of ~ 0.6 V \cdot cm. Open eye diagrams at 200 Gb/s PAM4 have been achieved with a swing voltage of 1.6 V and energy consumption of 6.3 fJ/bit. This work is pretty nice and the following comments should be addressed before it is to be accepted.

1. Could the authors show the measurement results for more samples of the same batch, which is important to show the uniformity of the fabrication.

Response: We are grateful for the suggestion. The uniformity is important to show the quality of the fabrication and device design. We have measured PAM4 eye diagrams of this two-segment Z-shape MRM across six different dies. Without the device-specific calibration, all six MRMs exhibit open 180 Gb/s PAM4 eye diagrams using the same driving signals, as shown in Fig. S7. The demonstration of 180 Gb/s eyes here is because the upper limit of the setup is 200 Gb/s (the arbitrary waveform generator can only generate frequency content up to 50 GHz), achieving 200 Gb/s PAM4 eye requires careful calibration and precise tuning of LSB/MSB time delay for each device. The die within the red block is the one measured and reported in the main manuscript, and the six surrounding dies show similar performance. This Z-shape design is realized through four standard implantations rather than the shaded implantations, making it well-suited for mass production. The exceptional uniformity of this design paves the way for the commercialization of high-speed, efficient two-segment Z-shaped MRMs, enabling their widespread deployment.

Figure S7. Measured 180 Gb/s PAM4 eye diagrams of the two-segment Z-shape MRMs on different dies.

To address your suggestion, we have added “The uniformity of the Si MRM chips is shown in Supplementary Information V.” in the **Fabrication** section;

and added section **Uniformity** in **Supplementary Information** as “The device uniformity is another important figure of merit. To demonstrate the consistent performance of the Z-shape MRM design, PAM4 eye diagrams were measured across six different dies. As depicted in Fig. S7, all of these MRMs exhibit open 180 Gb/s PAM4 eye diagrams using the same driving signals without device-specific calibration. The demonstration of 180 Gb/s eyes here is because the upper limit of the setup is 200 Gb/s (as detailed in VI. Experimental setup for eye measurement), achieving 200 Gb/s PAM4 eye requires careful calibration and precise tuning of LSB/MSB time delay for each device. The die within the red block is the one measured and reported in the main manuscript, and the six surrounding dies show similar performance. This Z-shape design is realized through four standard implantations rather than the shaded implantations, making it well-suited for mass production. The exceptional uniformity of this design paves the way for the commercialization of high-speed, efficient two-segment Z-shaped MRMs, enabling their widespread deployment.”.

2. As the authors mentioned, the spacing between channels 1 and 2 is 0.76 nm, which is actually 66.6% of the desired channel spacing of 1.14 nm. Therefore, the claim of “without heater tuning, channel 1 through channel 5 are already uniformly distributed within one FSR” is convinced and should be modified.

Response: Thank you for your thorough revision. You are correct, the spacing between channels 1 and 2 is narrower than the ideal channel spacing due to the fabrication errors. Our statement is not rigorous. We have modified this sentence into: “The measured transmission spectrum of the 5-channel MRM array at zero bias is depicted in Fig. 5(b), channel 1 through channel 5 are approximately evenly distributed within one FSR without heater tuning.” to **DWDM MRMs** section on page 6.

3. The bending radius is chosen as 12 μm and the corresponding FSR is about 5.7 nm, which is quite small if more channels are involved. How much is the bending loss for the present design? Could the radius be reduced? Please give a comment on this.

Response: Thank you for the insightful comment. A smaller radius is desired for wider FSR and more channels. Fortunately, the current Silicon photonics foundries can provide mature processes to achieve small bending loss. Our wafer was taped out at Advanced Micro Foundry (AMF), where the 5 μm is the typical radius of bend, and the 90° 5 μm -radius bend only has a loss of ~ 0.015 dB. The current 12 μm radius exhibits negligible bend loss. If we reduced the bend radius to 5 μm , the FSR of the microring will increase to ~ 13.7 nm (i.e., ~ 2.4 THz). Therefore, 12 channels can be supported with the same 200 GHz channel spacing. It will result in about 0.06 dB bend loss in total, but it is way less than the free carrier absorption loss from the silicon PN junction and won't affect the modulator performance. To address this, we have added

“The DWDM structure provides scalability for ever-increasing data rates. By reducing the MRM radius to 5 μm , the FSR of the MRM can be extended to ~ 13.7 nm with a bend loss of only about 0.06 dB. With the same channel spacing, 12 channels can be supported, easily scaling up the full data rate to 2.4 THz.” to the **Discussion** section on page 8.

4. Enlarged views for the MRM and the MRM array should be provided in Fig. 4(a) and Fig. 5(a).

Response: Thank you for the great suggestion, enlarged views for the MRM can help illustrate the structure of the device. Since the MRM and the MRM array share the same device structure, we added the enlarged SEM picture of the microring structure in Fig. 4(a). The modified figure is shown as below: “

Figure 4. (a) Micrograph of a two-segment Si MRM. Measured eye diagrams of 100 Gb/s NRZ of (b) LSB, (c) MSB, (d) both LSB and MSB; and (e) 200 Gb/s PAM4 of both LSB and MSB.”

5. What kind of laser source used in the experiment with the MRM array? Is it a multi-wavelength laser?

Response: That is a great question. The current laser source is a commercially available tunable laser, the Santec TSL 510, where the input laser wavelength can be tuned for different channels. Our research group has demonstrated comb lasers with the same 200 GHz channel spacing, but

they are currently located on a different wafer and link testing is hampered by high loss of its grating coupler. In the future, we will integrate the comb laser and DWDM microring modulator array on the same chip and use the comb laser as the optical source. To clarify it, we have modified the **Eye diagrams measurements** section “A commercial tunable CW laser, Santec TSL 510, was used as the input optical source”.

6. The authors claimed “this DWDM Si MRM array demonstrates remarkably small crosstalk, even without any thermal tuning, it exhibits < -33 dB crosstalk between the two closest channels. This value can be further improved to ~ -57 dB level by tuning the channel spacing to the ideal 1.14 nm.” Please give more details to explain the crosstalk (Fig. 5h) from which to which. Is it optical or electrical crosstalk?

Response: Sorry for the confusion. Figure 5(h) shows the measured EO responses of the channel 2 microring modulator at different wavelengths. The orange line is the measured EO response at the channel 2 desired wavelength, $\lambda_{C2} = 1306.19\text{nm}$. Therefore, it shows a flat and strong response with a high 3dB bandwidth. Meanwhile, the blue and green lines are the EO response of the channel 2 modulator at neighboring channel input wavelengths. It will show us the optical crosstalk of the DWDM microring array. The reason we chose channel 2 is because channel 2 exhibits the narrowest channel spacing from its neighboring channels, therefore, it would tell us the worst optical crosstalk of the entire array. As Fig. 5(h) shows, the blue line is measured with channel 1 desired wavelength, $\lambda_{C1} = 1305.43\text{nm}$. Even with the narrowest channel spacing, $\Delta\lambda \sim 0.76\text{nm}$, the blue line achieves a low EO response. On average, the optical crosstalk between channel 1 and channel 2 is less than -33 dB. On the other hand, the channel spacing between channel 2 and channel 3 is wider, which is about 1.13nm, aligning well with the ideal 200 GHz channel spacing. As a result, the green line exhibits a lower EO response, which is about -57.7dB.

Figure 5. (h) Measured channel crosstalk of the DWDM MRM array.

To clarify this, we have modified Fig. 5(h) with a new Y-axis caption “C2 MRM EO Response (dBe)” and new legends for the measured lines. Additionally, we have modified the crosstalk paragraph “The channel crosstalk is another pivotal figure of merit for the DWDM modulators.”

In order to quantify the optical crosstalk, the EO response ($|S_{21}|^2$) of an MRM was measured at adjacent and its own channel wavelengths. Channel 2 MRM is chosen here because it has the narrowest channel spacing from its neighboring channels. The measured channel crosstalk is illustrated in Fig. 5(h), the orange line is the measured EO response of the channel 2 MRM at its own wavelength λ_{C2} , displaying the anticipated flat, strong response and high 3 dB bandwidth; the blue line represents the channel 2 MRM response at λ_{C1} ; and the green line is the response of the channel 2 MRM at λ_{C3} . The wavelength difference between channels 1 and 2 is $1306.19 - 1305.43 = 0.76$ nm, which is slightly less than the ideal channel spacing of 1.14 nm (~ 200 GHz) due to the fabrication variance. This narrower spacing is also evident in the transmission spectrum in Fig. 5(b). Despite the 0.76 nm channel spacing, on average, the channel 1 wavelength exhibits ~ -33.4 dB smaller EO response than that of channel 2. On the other hand, the wavelength difference between channels 2 and 3 is $1307.32 - 1306.19 = 1.13$ nm, aligning closely with the ideal channel spacing. Consequently, the channel 3 wavelength has an even weaker EO response, which is ~ -57.7 dB less than the actual channel response. The increased EO responses at high frequencies for channel 1 and 3 wavelengths are due to the optical peaking effect from the wavelength detuning. In short, this DWDM Si MRM array demonstrates remarkably small crosstalk, even without any thermal tuning, it exhibits < -33 dB crosstalk between the two closest channels. This value can be further improved to ~ -57 dB level by tuning the channel spacing to the ideal 1.14 nm. Furthermore, the characterization of large-signal crosstalk was conducted by simultaneously operating channel 2 MSB and channel 3 LSB, as illustrated in Supplementary Information IV. At a data rate of 100 Gbaud/s, the eye quality remains unaffected, affirming the negligible impact of channel crosstalk.” on page 7.

We have also added the section **Linearity and crosstalk** in **Supplementary Information** to further explore the crosstalk performance: “The segment length ratio of $\sim 2:1$ makes the MRM an optical DAC to simplify the pulse amplitude modulation with four levels (PAM4) driving signal to two non-return-to-zero (NRZ) driving signals. The equally spaced four levels are critical to realize PAM4 modulation. Figure S4(a) illustrates the measured optical transmission spectrum of the MRM at four bias levels: 1) LSB = 0 V, MSB = 0 V; 2) LSB = -4 V, MSB = 0 V; 3) LSB = 0 V, MSB = -4 V; and 4) LSB = -4 V, MSB = -4V. The resonant wavelength of the MRM red shifts with bias voltage levels. To quantify the linearity of the optical DAC, the measured spectrum on a linear scale is also plotted as shown in Fig. S4(b). At a fixed laser wavelength indicated by the black dash line, the optical power differences of the four bias levels are about 0.16, 0.18, and 0.16, respectively, with a maximum spacing deviation of $\sim 8\%$. This two-segment optical DAC has good linearity and enables nearly equally spaced PAM4 modulation.

Figure S4. Measured transmission spectrum of the two-segment MRM at four bias levels on (a) dB scale and (b) linear scale.

In addition to measuring optical transmission intervals at DC bias voltages, the RF responses of LSB, MSB, and their combination were also conducted to demonstrate the linearity in large-signal RF cases. Figure S5 displays the captured waveforms on the digital communication analyzer (DCA) at 50 Gbaud/s data rate, where the pink waveform indicates driving only LSB, the yellow waveform corresponds to driving only MSB, and the blue curve represents the simultaneous driving of both LSB and MSB. If there is no time offset between the driving signals of LSB and MSB, all three waveforms should exhibit similar shapes. As shown in Fig. S5(a), three curves share similar waveforms. Moreover, the optical modulation amplitude (OMA) ratio of the three waveforms is approximately 1:2:3, underscoring great linearity in the RF domain. It is worth noting that the driving signals and swing voltages are identical for both LSB and MSB. An advantage of the two-segment design is the ability to independently tune the two driving signals, allowing for the possibility of achieving even better linearity by making slight adjustments to the amplitudes of the driving signals. On the other hand, by introducing an integer-bits offset between the driving signals of LSB and MSB, the combined response should exhibit four distinct levels of modulation. Figure S5(b) presents the captured waveforms with a 20 ps offset, i.e., 1 bit, and an example six-bit sequence is depicted by the red dash lines. The LSB waveform shows a sequence of [0, 1, 1, 0, 1, 0], whereas the MSB waveform is 1 bit later with a sequence of [0, 0, 1, 1, 0, 1]. The combined waveform of both LSB and MSB displays a sequence of [0, 1, 3, 2, 1, 2], aligning with the requirements of the optical DAC. This result not only demonstrates the good linearity of the large-signal RF response but also indicates the absence of noticeable electrical crosstalk.

Figure S5. Measured waveforms of LSB, MSB, and LSB + MSB at 50 Gbaud/s data rate: (a) No offset between LSB and MSB drive signals. (b) 20 ps offset (i.e., 1 bit) between LSB and MSB drive signals.

Due to the limited channel number in the current setup, we can only provide two high-speed driving signals to this MRM array. In order to closely simulate the real product operating condition, which involves driving multiple channels simultaneously using custom complementary metal-oxide-semiconductor (CMOS) driver circuits, we shifted half a channel to evaluate optical crosstalk. The micrograph of the measured MRM array is shown in Fig. S6(a), the customized probe simultaneously probes channel 2 MSB and channel 3 LSB, rather than probing LSB and MSB of one channel. Figure S6(b) and (c) show the measured 100 Gb/s NRZ eye diagrams of the channel 2 MSB. The input laser wavelength is aligned with channel 2, channel 2 MSB driving signal remains on, while the channel 3 LSB driving signal is off (left) and on (right). The signal-to-noise ratios (SNRs) of the two 100 Gb/s NRZ eyes are very close, around 3.1. Likewise, the 100 Gb/s NRZ eye diagrams of the channel 3 LSB are presented in Fig. S6 (d) and (e). Toggling the driving signal off and on of its neighboring channel 2 does not impact the eye quality, both eye diagrams have an SNR of ~ 2.9 . Hence, this measurement serves as compelling proof that the dense wavelength division multiplexing (DWDM) MRM array exhibits negligible optical crosstalk under 100 Gbaud/s modulation conditions. Consequently, it is entirely feasible to support a total data rate of 1 Tb/s with this DWDM MRM array.

Figure S6. (a) Micrograph of the probed MRM array for large signal crosstalk measurements. Measured 100 Gb/s NRZ eye diagrams: input wavelength is λ_{C2} , C2 driving signal is on, C3 driving signal is (b) off and (c) on; input wavelength is λ_{C3} , C3 driving signal is on, C2 driving signal is (d) off and (e) on.”

7. How about the real loss of the MRR waveguide in experiment?

Response: Thank you for your great comment. The real loss of the MRR waveguide is an important parameter, which can be calculated from the measured transmission spectrum. As shown in Fig. S1(a), the measured finesse of the MRR is $F = FSR/FWHM \approx 16$ and the ER of the MRR is ~ 16 dB. They can also be expressed by

$$F \approx \frac{2\pi}{\delta_k + \delta_r}, \quad (1)$$

$$ER \approx \left| \frac{\delta_k + \delta_r}{\delta_k - \delta_r} \right|^2, \quad (2)$$

where δ_k is the coupling loss coefficient and δ_r is the ring waveguide propagation loss coefficient. Based on the measured values of F and ER , these two coefficients can be extracted: $\delta_k \sim 0.167$ and $\delta_r \sim 0.229$. The coupling coefficient value is also consistent with the FDTD simulation results of the coupling region. The transmission spectrum can then be plotted by using

$$T = \left| \frac{t - ae^{-j\phi}}{1 - tae^{-j\phi}} \right|^2, \quad (3)$$

where t is the field transmission of the coupler, a is the roundtrip field transmission of the MRR, and ϕ is the roundtrip phase of the MRR. The relationship between the field transmission and loss coefficient is $t^2 = \exp(-\delta_\kappa)$ and $a^2 = \exp(-\delta_r)$. The transmission spectrum calculated according to Eq. 3 is shown in Fig. S1(b), which is in good agreement with the measured transmission spectrum. Since $a^2 = \exp(-\delta_r) \approx 0.795$, the actual loss of the MRR is ~ 0.205 (i.e., ~ -1 dB) and the MRR propagation loss per unit length is -133 dB/cm. This value is close to the simulated propagation loss from Lumerical -129 dB/cm.

Figure S1. (a) Measured and (b) calculated transmission spectrum of a single-channel MRM.

We have added “MRM waveguide losses are analyzed in Supplementary Information I.” on page 3;

and added the section **Microring waveguide loss** in **Supplementary Information** as

“The waveguide loss of the microring can be extracted from the measured transmission spectrum, shown in Fig. S1 (a). The measured finesse of the microring modulator (MRM) is \$F = FSR/FWHM \sim 16\$ and the extinction ratio is \$ER \sim 16\$ dB. Both parameters can be expressed using the MRM loss coefficients by

$$F \approx \frac{2\pi}{\delta_\kappa + \delta_r}, \quad (1)$$

$$ER \approx \left| \frac{\delta_\kappa + \delta_r}{\delta_\kappa - \delta_r} \right|^2, \quad (2)$$

where δ_κ is the coupling loss coefficient and δ_r is the microring waveguide propagation loss coefficient. Based on the measured values of F and ER , loss coefficients can be calculated, $\delta_\kappa \sim 0.167$ and $\delta_r \sim 0.229$. The coupling coefficient value is also consistent with the coupler simulation results using Lumerical FDTD. Hence, the transmission spectrum can be plotted by

$$T = \left| \frac{t - ae^{-j\phi}}{1 - tae^{-j\phi}} \right|^2, \quad (3)$$

where t is the field transmission of the coupler, a is roundtrip field transmission of the microring, and ϕ is the roundtrip phase of the microring. The relationship between the field transmission and loss coefficients is $t^2 = e^{-\delta\kappa}$ and $a^2 = e^{-\delta r}$. The calculated transmission spectrum according to Eq. 3 is shown in Fig. S1(b), which is in good agreement with the measured transmission spectrum. Since $a^2 = e^{-\delta r} \sim 0.795$, the actual loss of the MRR is ~ 0.205 (i.e., ~ -1 dB), and the MRR propagation loss per unit length is -133 dB/cm. This value is close to the simulated propagation loss of -129 dB/cm from Lumerical Charge and Mode.”

8. The eye-diagram quality seems not very high. Please give an explanation and a discussion on how to improve it.

Response: We are very grateful for the suggestion. The eye diagram quality is mainly limited by the test setup. The 3dB bandwidth of the electrical power amplifiers, bias-tees, optical receiver, and RF cables are 60 GHz, 50 GHz, 65 GHz, and 65 GHz, respectively. The combination of these components results in a loss of ~ 10 dB in 50 GHz. In fact, we first measured the eye diagram using 50 GHz power amplifiers, and the highest data rate we could achieve was about 160 Gb/s. Therefore, we purchased 60 GHz power amplifiers to achieve 200 Gb/s. Besides that, according to the M8194A data sheet, the 120 GSa/s arbitrary waveform generator (AWG) can generate frequency content up to 50 GHz. The 100 Gb/s NRZ signals are the upper limit of the AWG, our MRMs have pushed the experimental setup to its limitation. A higher speed experimental setup can significantly improve our eye diagram quality.

To be clearer and in accordance with the reviewer's comments, we have added “It is worth noting that 200 Gb/s eyes are the upper limit of the experimental setup, higher bandwidth setup can further improve the data rate.” to the **DWDM MRMs** section on page 7;

modified the Properties Comparison Table 1 as “

Ref.	Junction type	Modulation structure	Radius (μm)	Q	Bandwidth (GHz)	$V_\pi \cdot L$ (V·cm)	Operating λ (nm)	Data rate (Gb/s)	Energy cost (fJ/bit)
14	vertical	one-segment	4.8	6800	21	-	1550	1×44 (NRZ)	0.9
15	vertical	one-segment	5	6780	24	-	1310	1×40 (NRZ)	4
16	vertical	one-segment	5	4800	35	0.37 [†]	1310	1×100 (PAM4)	-
17	vertical	one-segment	8	5600	67 (-3dB IL)	0.8	1310	1×200 (PAM4)	-
18	lateral	one-segment	3	1050	67	0.94	1550	1×200 (PAM4)	-
19	lateral	one-segment	8	4500	52	0.825	1310	1×240 (PAM8)	-
20	lateral	two-segment	12	7500	18	-	1310	1×40 (PAM4)	-
21	lateral	two-segment	12	3600	40	1.0	1310	1×100 (PAM4)	9.9
22	L-shape	one-segment	10	5000	50	0.52	1310	1×128 (PAM4)	18
23	L-shape	one-segment	4	4000	54	0.53	1310	1×240 (PAM4)	-
30	U-shape	one-segment	32.5	26200	13.5	0.46	1310	1×13 (NRZ)	-
31	Z-shape	one-segment	9	7000	25 (-6dB IL) 42 (-3dB IL)	0.92	1310	1×100 (PAM4)	14
this work	Z-shape	two-segment	12	3700	48.6 (-6dB IL) 58.1 (-3dB IL)	0.6	1310	5×200* (PAM4)	6.3

[†] 0.37 V·cm is measured from 0.5 V to -0.5 V; $V_\pi \cdot L$ would increase to > 0.53 V·cm in the higher reverse voltage region.

* 200 Gb/s limited by the bandwidth of the experimental setup.

Table 1. Properties of depletion mode MRMs based on different types of Si p-n junctions.

”;

—;

as well as modified the **Eye diagrams measurements** section “The eye diagrams of the MRMs have been performed with a 120 GSa/s arbitrary waveform generator (AWG) M8194A, which can generate frequency content up to 50 GHz. It provides two PRBS9 NRZ signals for the LSB and MSB. A signal generator was used to generate a 2.5 GHz clock signal for the AWG clock reference, the digital communication analyzer (DCA) trigger reference, and the DCA precision reference. The highest swing voltage V_{pp} of the AWG is 0.8 V, therefore two identical 60 GHz electrical power amplifiers were added to amplify the swing voltage. After that, two identical 50 GHz bias-tees were connected to combine the RF signals and DC bias voltage. The measured V_{pp} after the bias-tee at 100 Gb/s NRZ, i.e. the LSB and MSB driving voltage, is 1.6 V. A commercial tunable CW laser, Santec TSL 510, was used as the input optical source, and the MRM output light was amplified by a Praseodymium-doped fiber amplifier (PDFA) to compensate for the link losses. A 65 GHz optical module N1030A was used to receive the modulated signals on the DCA. The measurement setup is illustrated in Supplementary Information VI. The signal distortion caused by the power amplifier, bias-tee, and RF cables was calibrated by the AWG internal calibration tool. In order to eliminate the amplified spontaneous emission (ASE) noise from the PDFA, the received patterns were averaged 64 times. A more than half-baud rate 4th-order Bessel filter and a 21-tap FFE were used at the receiver. The TDECQ values of the 200 Gb/s PAM4 eye diagrams were measured at an SD-FEC threshold of SER at $1E-2$. The 200 Gb/s PAM4 is the upper limit of the current eye diagram experimental setup, a higher data rate is achievable with higher bandwidth AWG and RF components.”.

Reviewer #2

In this manuscript, the authors demonstrated a 1 Tbps DWDM microring modulator silicon chip empowered by two-segment z-shape junctions. Each silicon depletion-mode modulator achieved a bandwidth of ~ 48.6 GHz and a modulation efficiency with $V\pi \cdot L$ of ~ 0.6 V \cdot cm. Although a two-segment Z-Shape junction is introduced, and a dense wavelength division multiplexing microring modulator array is demonstrated, I do not recommend this manuscript to be accepted in Nature Communications under its current version due to the following reasons.

1. In general, the innovation of this article is insufficient. The optical DAC, Z-shape, and DWDM microring modulator array have been reported in other papers, and the performance of a single depletion-mode microring modulator is close to or even slightly better than that of this paper, as shown in Table 1.

Response: We are very grateful for your valuable comment. Compared to the Z-shape p-n junction in Ref. [31], our design has distinctive features. The left side of the Figure below shows the carrier concentration profiles of the junction in Ref. [31]. This Z-shape junction is implemented similarly to the L-shape junction, with one n-type doping concentration and one p-type doping concentration in the microring waveguide. Although the Z-shaped junction increases the carrier-light interaction area, the trade-off between modulation efficiency and bandwidth becomes worse due to the increased junction capacitance. Ref. [31] does not address this problem. Differently, our structure utilizes four separate dopant implants, which are fully compatible with Si photonics foundries as part of the standard process, making it readily fabricable. This design provides an additional degree of freedom to independently adjust the doping concentrations in the slab region and the waveguide core region. Therefore, high modulation efficiency, reasonable free-carrier absorption, and small series resistance can be achieved simultaneously. The small series resistance effectively alleviates the trade-off between modulation efficiency and bandwidth. Meanwhile, we have demonstrated for the first time a Z-shape junction MRM with a two-segment structure to reduce the junction capacitance as well. It further eases the trade-off, as we shown in the manuscript, two-segment structure can improve the junction RC-limited bandwidth by $> 20\%$. Due to the above innovations, the bandwidth of our MZM is not compromised by the Z-shape junction design. The performance of our Z-shape MRM is much better than the Z-shape MRM in Ref. [31]. In comparison, at 1310 nm, our MRM exhibits a $\sim 94\%$ improvement in bandwidth at insertion loss of -6 dB, a $\sim 53\%$ enhancement in modulation efficiency, and doubling the eye diagram data rate.

Figure. **Left (Ref [31]):** The carrier concentration profiles of the Z-shape PN junction at a bias voltage of (a) 0 V and (b) -2 V. **Right (this work):** (b) Cross-sectional diagram of the Z-shape junction with its corresponding (c) waveguide TE mode and (d) electric field at -3 V.

You are correct that there are several single depletion-mode MRMs that perform close to or even slightly better than our Z-shaped MRM. The designed Z-shaped p-n junction was expected to provide greater overlap compared to other junctions, but the experimental results did not fully reflect this. This difference can be attributed to the actual doping profile of the Z-junction, which may differ from the original design because we simulated the implant dose and energy while the devices were fabricated at AMF. The actual doping profile is affected by many factors in the fabrication, such as the thickness of implantation mask, dopant activation temperature, and so on. Nevertheless, as the first batch of Z-shape MRMs, its modulation efficiency is already close to that of other state-of-the-art MRMs. Further optimization of the Z-shape doping profile holds the potential for achieving higher modulation efficiency $V_{\pi} \cdot L < 0.5 \text{ V} \cdot \text{cm}$.

In accordance with the reviewer's comments, we have modified “Here we propose a Si MRM featuring two-segment Z-shape junctions to alleviate this trade-off. The Z-shape junction profile increases the effective carrier-light interaction and reduces the device series resistance, meanwhile the two separate junctions design reduces the junction capacitance and facilitates pulse amplitude modulation with four levels (PAM4) using two non-return-to-zero (NRZ) signals” in **Introduction** section;

modified the comparison table: “

Ref.	Junction type	Modulation structure	Radius (μm)	Q	Bandwidth (GHz)	$V_{\pi} \cdot L$ (V·cm)	Operating λ (nm)	Data rate (Gb/s)	Energy cost (fJ/bit)
14	vertical	one-segment	4.8	6800	21	-	1550	1×44 (NRZ)	0.9
15	vertical	one-segment	5	6780	24	-	1310	1×40 (NRZ)	4
16	vertical	one-segment	5	4800	35	0.37 [†]	1310	1×100 (PAM4)	-
17	vertical	one-segment	8	5600	67 (-3dB IL)	0.8	1310	1×200 (PAM4)	-
18	lateral	one-segment	3	1050	67	0.94	1550	1×200 (PAM4)	-
19	lateral	one-segment	8	4500	52	0.825	1310	1×240 (PAM8)	-
20	lateral	two-segment	12	7500	18	-	1310	1×40 (PAM4)	-
21	lateral	two-segment	12	3600	40	1.0	1310	1×100 (PAM4)	9.9
22	L-shape	one-segment	10	5000	50	0.52	1310	1×128 (PAM4)	18
23	L-shape	one-segment	4	4000	54	0.53	1310	1×240 (PAM4)	-
30	U-shape	one-segment	32.5	26200	13.5	0.46	1310	1×13 (NRZ)	-
31	Z-shape	one-segment	9	7000	25 (-6dB IL) 42 (-3dB IL)	0.92	1310	1×100 (PAM4)	14
this work	Z-shape	two-segment	12	3700	48.6 (-6dB IL) 58.1 (-3dB IL)	0.6	1310	5×200* (PAM4)	6.3

[†] 0.37 V·cm is measured from 0.5 V to -0.5 V; $V_{\pi} \cdot L$ would increase to $> 0.53 \text{ V} \cdot \text{cm}$ in the higher reverse voltage region.

* 200 Gb/s limited by the bandwidth of the experimental setup.

Table 1. Properties of depletion mode MRMs based on different types of Si p-n junctions.

”;

as well as rewritten the **Discussion** section as “In this work, a Si MRM with two-segment Z-shape junctions has been demonstrated for use in the next-generation optical interconnects. The properties comparison of depletion mode MRMs with diverse Si p-n junctions, including vertical, lateral, L-shape, U-shape, and Z-shape junctions, are summarized in Table 1. To enhance modulation efficiency, longer depletion regions have been adopted for better mode overlap, such as the vertical, L-shape, and U-shape p-n junctions. The U-shape junction offers the best

modulation efficiency $V\pi\cdot L$ of 0.46 V·cm. However, it also results in the longest junction capacitor, nearly twice as long as the L-shape and Z-shape junctions. The ensuing large RC time constant will hinder the high-speed operation. One potential solution to mitigate this RC issue is to lower the doping concentrations in the U-shape junction, allowing the middle p-type layer to be almost entirely depleted, thereby significantly reducing the junction capacitance³⁰. Unfortunately, the low doping concentrations result in a much higher Q factor, which in turn restricts the photon lifetime-limited bandwidth. Consequently, the U-shape design is more suitable for low-speed, high-efficiency modulation. Other than that, L-shape junctions exhibit the second-best modulation efficiency²². The simulated Z-shape junction is expected to offer ~11% better overlap compared to L-shape junctions, but the experimental results do not fully reflect this. This discrepancy can be attributed to the fact that the actual fabricated Z-shaped doping profile deviates from the ideal design. As the first batch of Z-shape MRMs, its modulation efficiency is already close to that of other state-of-the-art MRMs. Further optimization of the Z-shape doping profile holds the potential for achieving higher modulation efficiency $V\pi\cdot L < 0.5$ V·cm. Besides the enhanced overlap between the waveguide mode and the junction depletion region, our innovative design offers high bandwidth at the same time. The 4 distinct implantations create higher doping concentrations inside the slab region than the waveguide core region to achieve a reduced series resistance without leading to high free carrier absorption loss. With this unique degree of freedom, our Z-shape MRM exhibits a similarly high bandwidth even with larger microring radius. Meanwhile, the two-segment structure is employed to further reduce the capacitance of the Z-shape junctions. Therefore, this MRM combines a high RC-limited bandwidth with better modulation efficiency. In addition, the two-segment architecture simplifies the complex PAM4 driving signals into two NRZ signals. With the adoption of this optical DAC, the MRM can modulate higher-speed data with better linearity and lower CMOS driver energy consumption.

Thanks to the innovative design, our two-segment Z-shape MRM exhibits a high 3dB bandwidth of ~ 48.6 GHz at the wavelength with maximum modulation slope, a large modulation efficiency with $V\pi\cdot L$ of ~ 0.6 V·cm, and thus supports open 200 Gb/s PAM4 eye diagrams with only 1.6 V V_{pp} and 6.3 fJ/bit energy consumption. The demonstrated MRM achieves state-of-the-art performance for Si depletion mode modulators. Compared to the previous leading two-segment MRM with a lateral junction²¹, it enables a ~ 21 % increase in bandwidth, a remarkable ~ 67 % enhancement in modulation efficiency, a doubling of the data rate, and a ~ 36% reduction in energy cost per bit. With this high performance, the 5-channel DWDM Si MRM array based on two-segment Z-shape junctions has been realized, which supports 1 Tb/s (5×200 Gb/s PAM4) data rates. The inherent offset resonances of the microrings allow the DWDM MRM array to act as an effective multiplexer (MUX), achieving an impressively low crosstalk of < -33 dB directly, with the potential for further improvement to ~ -57 dB with control circuits. The DWDM structure provides scalability for ever-increasing data rates. By reducing the MRM radius to 5 μm , the FSR of the MRM can be extended to ~ 13.7 nm with a bend loss of only about 0.06 dB. With the same channel spacing, 12 channels can be supported, easily scaling up the full data rate to 2.4 THz. To encapsulate, this work showcases a total of 1 Tb/s DWDM modulation integrated on a Si chip with a small V_{pp} of 1.6 V, paving the way to satisfy the ever-increasing data demands for intrar- and inter-chip interconnects.”

[31] Cai, H., Fu, S., Yu, Y. & Zhang, X. Lateral-zigzag pn junction enabled high-efficiency silicon micro-ring modulator working at 100gb/s. *IEEE Photonics Technol. Lett.* 34, 525–528 (2022)

2. The article has mentioned many times that "the two segment structure is employed to reduce the capacitance of the Z-shape junctions ", what is the benefit of reducing the junction capacitance? The authors seem to want to express that the bandwidth can be improved, but from Figure 3, the junction capacitance of the MSB is larger than that of the LSB, yet the measured bandwidth in both segments is almost the same.

Response: Thank you for the great question, the reduced capacitance is one of the key advantages of the two-segment structure. Z-shape p-n junction offers better overlap between the depletion region and optical mode, however, it leads to a larger junction capacitance. The trade-off between the modulation efficiency and RC time-limited bandwidth hinders the MRM performance. The two-segment structure helps alleviate this trade-off.

Figure 3. Measured and fitted Smith charts of S11 for (a) LSB and (d) MSB at -3 V. Fitted equivalent circuits and corresponding junction RC responses for (b) LSB and (e) MSB at -3 V. Measured EO responses ($|S_{21}|^2$) for (c) LSB and (f) MSB with detuning of $\Delta\lambda \sim 0.1$ nm at -3 V.

Figure 3 (a) and (d) show the measured S11 charts for the LSB and MSB at -3 V. The equivalent circuits for both segments can be derived through the S11, shown as insets in Fig. 3 (b) and (e). Although the product of the junction capacitance and series resistance, $C_j \cdot R_s$, is almost independent of the junction length, the smaller capacitance provides higher RC bandwidth considering the entire equivalent circuits. The LSB and MSB junctions have RC bandwidths of ~ 79.1 GHz and 64.5 GHz, respectively. As a comparison, the equivalent one-segment MRM would yield a much smaller RC bandwidth of ~ 53.6 GHz. The measured EO bandwidths of both segments are close to each other, that is because the RC bandwidths of both segments are higher than the photon lifetime-limited bandwidth $f_{ph} \sim 62$ GHz. The f_{ph} dominates the EO bandwidth, so that the overall EO bandwidth does not vary much, ranging from 48 to 49 GHz. Differently, if

the RC bandwidth is less than f_{ph} , it will exhibit a larger influence. For example, if it is a one-segment MRM with an RC bandwidth of ~ 53.6 GHz, the overall EO bandwidth will reduce to ~ 41 GHz. As a result, the two-segment design improves the RC bandwidth by $> 20\%$ and the consequent EO bandwidth by $> 17\%$.

To make it clear, we have added the section **Electro-optical response** in **Supplementary Information** as

“A Z-shape Si p-n junction has been adopted in the MRM to enhance the overlap between the carrier variance and optical mode. However, the increased depletion interface of the Z-shape junction leads to a larger junction capacitance, which ultimately limits the overall bandwidth of the MRM. To address this trade-off, four dopant implants were applied to the microring waveguide. This design provides an additional degree of freedom to independently adjust the doping concentrations in the slab region and the waveguide core region. Therefore, high modulation efficiency, reasonable free-carrier absorption, and small series resistance can be achieved simultaneously. The small series resistance effectively alleviates the trade-off between modulation efficiency and bandwidth. To attain the next-generation data rate of 200 Gb/s per lane, just reducing the series resistance is insufficient to meet the required electro-optical (EO) bandwidth. Fortunately, the optical digital-to-analog converter (DAC), the two-segment structure, can further improve the RC bandwidth. Figure 3 (a) and (d) show the measured S11 charts for the least significant bit (LSB) and the most significant bit (MSB) segments at -3 V. The equivalent circuits for both segments can be derived through the S11, shown as insets in Fig. 3 (b) and (e). Although the product of the junction capacitance and series resistance, $C_j \times R_s$, is almost independent of the junction length, the smaller capacitance still provides higher RC bandwidth considering the entire equivalent circuits. The LSB and MSB junctions have RC bandwidths of ~ 79.1 GHz and 64.5 GHz, respectively. As a comparison, the equivalent one-segment MRM would yield a much smaller RC bandwidth of ~ 53.6 GHz.

The measured EO bandwidths of both segments, as depicted in Fig. S2, are close to each other. This is because the RC bandwidths of both segments are higher than the photon lifetime-limited bandwidth $f_{ph} \sim 62$ GHz. The f_{ph} dominates the EO bandwidth, therefore the overall bandwidth does not vary much. Differently, if the RC bandwidth is less than f_{ph} , it will exhibit a larger influence. For instance, with the one-segment MRM featuring a 53.6 GHz RC bandwidth, the overall EO bandwidth will reduce to ~ 41 GHz. As a result, the two-segment design improves the RC bandwidth by $> 20\%$ and the consequent EO bandwidth by $> 17\%$.

Figure S2. Measured EO response ($|S_{21}|^2$) of the two-segment MRM at insertion loss of (a) -6 dB and (b) -3 dB.

The EO response of the MRM is also affected by wavelength detuning. The MRM can achieve an extended 3 dB bandwidth by tuning the wavelength away from the resonance. Figure S2(a) is the measured EO response of the LSB and MSB segments at the wavelength corresponding to the maximum modulation slope, where the wavelength detuning $\Delta\lambda$ is ~ 0.1 nm. This wavelength is located at the -6 dB insertion loss (IL) point on the transmission spectrum of the MRM. The LSB and MSB have 3 dB bandwidths of ~ 48.9 GHz and 48.3 GHz, respectively. By further detuning the wavelength towards the IL of -3 dB, a distinct optical peaking effect can be observed, as shown in Fig. S2(b). The measured results are limited to 50 GHz due to the bandwidth constraint of the vector network analyzer. The fitted curves indicate that the LSB exhibits an enhanced 3 dB bandwidth of ~ 59.1 GHz and the MSB attains a 3 dB bandwidth of ~ 57.1 GHz.

Thanks to the novel design, this MRM achieves state-of-the-art EO bandwidth with a relatively large radius of $12\ \mu\text{m}$. By further reducing the MRM radius, the smaller junction capacitance can enhance the EO bandwidth considering the entire equivalent circuit. If the radius reduces to $4\ \mu\text{m}$, the simulated RC responses of the equivalent circuits of the LSB and MSB are illustrated in Fig. S3 (a) and (b), respectively. As expected, the RC time-limited bandwidth of the LSB extends from ~ 79.1 GHz to ~ 94.3 GHz, and the MSB's RC bandwidth improves from ~ 64.5 GHz to ~ 86.3 GHz. Using the simplified EO bandwidth equation, $f_{est} = f_{RC}f_{ph} / \sqrt{f_{RC}^2 + f_{ph}^2}$, the estimated EO bandwidth of both LSB and MSB will be > 50 GHz without detuning peaking effect, which is comparable to Intel's $4\ \mu\text{m}$ -radius MRM.

Figure S3. Simulated RC response of the (a) LSB and (b) MSB of the two-segment MRM with a $4\ \mu\text{m}$ radius.”.

3. The optical DAC needs to pay attention to its linearity, but the electrical crosstalk and the linearity are not characterized in this paper.

Response: We are very grateful for your insightful suggestion. The linearity of the optical DAC is an important figure of merit, we apologize that we did not include it in the manuscript. We have added the relevant experiments based on your advice. The transmission spectrum of the two-segment MRM at four different DC bias levels were measured, which shows good linearity with nearly equal intervals. The 50 Gbaud/s waveforms of LSB, MSB, and their combination were also conducted to demonstrate the linearity in the RF domain. The optical modulation amplitude of the three waveforms is approximately 1:2:3, indicating great linearity. Besides that, we introduced a 1-bit offset between the driving signals of LSB and MSB, the combined response of this optical DAC shows clear four levels for PAM4 modulation. The electrical crosstalk is not noticeable in the measured waveform.

In addition to that, we also modulated two channels simultaneously to see the crosstalk in eye diagrams. Since only two high-speed driving signals are available in our experimental setup, we shifted half a channel to evaluate crosstalk, i.e., probing the channel 2 MSB and channel 3 LSB at the same time. The 100 Gb/s NRZ eye diagrams were measured for channel 2 MSB and channel 3 LSB, respectively. Toggling the driving signal off and on of its neighboring channel does not impact the signal-to-noise ratios (SNRs) of the eyes. This DWDM two-segment MZM array exhibits good linearity and very small channel crosstalk.

To address this comment, we have added “The linearity of the optical DAC structure is detailed in Supplementary Information IV.” on page 3;

and added the section **Linearity and crosstalk** in **Supplementary Information** as “The segment length ratio of \$\sim 2:1\$ makes the MRM an optical DAC to simplify the pulse amplitude modulation with four levels (PAM4) driving signal to two non-return-to-zero (NRZ) driving signals. The equally spaced four levels are critical to realize PAM4 modulation. Figure S4(a) illustrates the measured optical transmission spectrum of the MRM at four bias levels: 1) LSB = 0 V, MSB = 0 V; 2) LSB = -4 V, MSB = 0 V; 3) LSB = 0 V, MSB = -4 V; and 4) LSB = -4 V, MSB = -4V. The resonant wavelength of the MRM red shifts with bias voltage levels. To quantify the linearity of the optical DAC, the measured spectrum on a linear scale is also plotted as shown in Fig. S4(b). At a fixed laser wavelength indicated by the black dash line, the optical power differences of the four bias levels are about 0.16, 0.18, and 0.16, respectively, with a maximum spacing deviation of \$\sim 8\%\$. This two-segment optical DAC has good linearity and enables nearly equally spaced PAM4 modulation.”

Figure S4. Measured transmission spectrum of the two-segment MRM at four bias levels on (a) dB scale and (b) linear scale.

In addition to measuring optical transmission intervals at DC bias voltages, the RF responses of LSB, MSB, and their combination were also conducted to demonstrate the linearity in large-signal RF cases. Figure S5 displays the captured waveforms on the digital communication analyzer (DCA) at 50 Gbaud/s data rate, where the pink waveform indicates driving only LSB, the yellow waveform corresponds to driving only MSB, and the blue curve represents the simultaneous driving of both LSB and MSB. If there is no time offset between the driving signals of LSB and MSB, all three waveforms should exhibit similar shapes. As shown in Fig. S5(a), three curves share similar waveforms. Moreover, the optical modulation amplitude (OMA) ratio of the three waveforms is approximately 1:2:3, underscoring great linearity in the RF domain. It is worth noting that the driving signals and swing voltages are identical for both LSB and MSB. An advantage of the two-segment design is the ability to independently tune the two driving signals, allowing for the possibility of achieving even better linearity by making slight adjustments to the amplitudes of the driving signals. On the other hand, by introducing an integer-bits offset between the driving signals of LSB and MSB, the combined response should exhibit four distinct levels of modulation. Figure S5(b) presents the captured waveforms with a 20 ps offset, i.e., 1 bit, and an example six-bit sequence is depicted by the red dash lines. The LSB waveform shows a sequence of [0, 1, 1, 0, 1, 0], whereas the MSB waveform is 1 bit later with a sequence of [0, 0, 1, 1, 0, 1]. The combined waveform of both LSB and MSB displays a sequence of [0, 1, 3, 2, 1, 2], aligning with the requirements of the optical DAC. This result not only demonstrates the good linearity of the large-signal RF response but also indicates the absence of noticeable electrical crosstalk.

Figure S5. Measured waveforms of LSB, MSB, and LSB + MSB at 50 Gbaud/s data rate: (a) No offset between LSB and MSB drive signals. (b) 20 ps offset (i.e., 1 bit) between LSB and MSB drive signals.

Due to the limited channel number in the current setup, we can only provide two high-speed driving signals to this MRM array. In order to closely simulate the real product operating condition, which involves driving multiple channels simultaneously using custom complementary metal-oxide-semiconductor (CMOS) driver circuits, we shifted half a channel to evaluate optical crosstalk. The micrograph of the measured MRM array is shown in Fig. S6(a), the customized probe simultaneously probes channel 2 MSB and channel 3 LSB, rather than probing LSB and MSB of one channel. Figure S6(b) and (c) show the measured 100 Gb/s NRZ eye diagrams of the channel 2 MSB. The input laser wavelength is aligned with channel 2,

channel 2 MSB driving signal remains on, while the channel 3 LSB driving signal is off (left) and on (right). The signal-to-noise ratios (SNRs) of the two 100 Gb/s NRZ eyes are very close, around 3.1. Likewise, the 100 Gb/s NRZ eye diagrams of the channel 3 LSB are presented in Fig. S6 (d) and (e). Toggling the driving signal off and on of its neighboring channel 2 does not impact the eye quality, both eye diagrams have an SNR of ~ 2.9 . Hence, this measurement serves as compelling proof that the dense wavelength division multiplexing (DWDM) MRM array exhibits negligible optical crosstalk under 100 Gbaud/s modulation conditions. Consequently, it is entirely feasible to support a total data rate of 1 Tb/s with this DWDM MRM array.

Figure S6. (a) Micrograph of the probed MRM array for large signal crosstalk measurements. Measured 100 Gb/s NRZ eye diagrams: input wavelength is λ_{C2} , C2 driving signal is on, C3 driving signal is (b) off and (c) on; input wavelength is λ_{C3} , C3 driving signal is on, C2 driving signal is (d) off and (e) on.”

4. In the manuscript, the SER used in the measurement of TDECQ is $1e-2$. What is the basis for this? The current used in the industry is KP4 encoding, which has a maximum error correction capability of $2e-4$.

Response: Thank you for the professional comment. Due to the speed limitation of our eye diagram measurement setup (50GHz AWG and 50GHz bias-tees), 100 Gbaud/s is the highest symbol rate that the setup can support. We first attempted to measure TDECQ at a SER of $2e-4$, however, we were unable to obtain any TDECQ values due to the setup limitation. Fortunately,

the third generation of Forward Error Correction (FEC) utilizes Soft Decision (SD), which provides a higher net coding gain of about 11 dB and thus is closer to the ideal Shannon limit. The threshold for SD-FEC is about $1e-2$ to $3e-2$ [23,27,28], therefore, we reported the TDECQ values at SER of $1e-2$.

To be clearer and in accordance with the reviewer's concerns, we have modified the “The transmitter dispersion eye closure quaternary (TDECQ) is measured with a soft-decision forward error correction (SD-FEC) threshold of symbol error rate (SER) at $1E-2^{23, 27, 28}$.” on page 6;

and modified the **Eye diagrams measurements** section “The eye diagrams of the MRMs have been performed with a 120 GSa/s arbitrary waveform generator (AWG) M8194A, which can generate frequency content up to 50 GHz. It provides two PRBS9 NRZ signals for the LSB and MSB. A signal generator was used to generate a 2.5 GHz clock signal for the AWG clock reference, the digital communication analyzer (DCA) trigger reference, and the DCA precision reference. The highest swing voltage V_{pp} of the AWG is 0.8 V, therefore two identical 60 GHz electrical power amplifiers were added to amplify the swing voltage. After that, two identical 50 GHz bias-tees were connected to combine the RF signals and DC bias voltage. The measured V_{pp} after the bias-tee at 100 Gb/s NRZ, i.e. the LSB and MSB driving voltage, is 1.6 V. A commercial tunable CW laser, Santec TSL 510, was used as the input optical source, and the MRM output light was amplified by a Praseodymium-doped fiber amplifier (PDFA) to compensate for the link losses. A 65 GHz optical module N1030A was used to receive the modulated signals on the DCA. The measurement setup is illustrated in Supplementary Information VI. The signal distortion caused by the power amplifier, bias-tee, and RF cables was calibrated by the AWG internal calibration tool. In order to eliminate the amplified spontaneous emission (ASE) noise from the PDFA, the received patterns were averaged 64 times. A more than half-baud rate 4th-order Bessel filter and a 21-tap FFE were used at the receiver. The TDECQ values of the 200 Gb/s PAM4 eye diagrams were measured at an SD-FEC threshold of SER at $1E-2$. The 200 Gb/s PAM4 is the upper limit of the current eye diagram experimental setup, a higher data rate is achievable with higher bandwidth AWG and RF components.”.

[23] Sakib, M. et al. A 240 gb/s pam4 silicon micro-ring optical modulator. In *2022 Optical Fiber Communications Conference and Exhibition (OFC)*, 01–03 (IEEE, 2022).

[27] Onohara, K. et al. Soft-decision-based forward error correction for 100 gb/s transport systems. *IEEE J. Sel. Top. Quantum Electron.* **16**, 1258–1267 (2010).

[28] Agrell, E. & Secondini, M. Information-theoretic tools for optical communications engineers. In *2018 IEEE Photonics Conference (IPC)*, 1–5 (IEEE, 2018).

Reviewer #3

This paper reports a silicon micro-ring modulator (MRM)-based optical transmitter chip capable of supporting 1 Tbps data transmission. The silicon MRM features a Z-shaped p-n junction with two segments designed for PAM4 modulation at 200 Gbps. The paper demonstrates an array of 5 such MRMs in series, forming a WDM transmitter with a projected capacity of 1 Tbps for data transmission.

The paper is well-written, and it provides a detailed description of the MRM and its characterization. However, there are several concerns that need to be addressed for this paper to meet the expected quality standards of Nature Communications.

1. The Z-shaped p-n junction is anticipated to offer greater overlap between the optical mode and the depletion region, potentially resulting in higher modulation phase efficiency when compared to other junction shapes, as shown in Fig. 2. However, the experimental measurements do not align with the simulation results. Specifically, the measured phase efficiency of $0.6 \text{ V}\cdot\text{cm}$ falls short of that achieved by the vertical junction at $0.37 \text{ V}\cdot\text{cm}$ (ref. 16), the L-shaped junction at $0.52 \text{ V}\cdot\text{cm}$ (ref. 22), or the U-shaped junction at $0.46 \text{ V}\cdot\text{cm}$ (<https://doi.org/10.1364/OE.25.008425>), which was not cited. A comprehensive discussion is essential to help readers comprehend the disparities between experimental results and theoretical predictions, as well as the technical advancements compared to prior research.

Response: We are very grateful for your professional comment. Indeed, the Z-shaped p-n junction was expected to offer a larger overlap compared to other junctions, but the experimental results do not fully reflect this. The discrepancy can be attributed to the actual doping profile of the Z-shape junction. The junction interface may deviate from the original design because the implantation dose and energy were simulated by us while the devices were fabricated at AMF. These discrepancies can arise from various aspects of the fabrication process, such as the thickness of the implantation mask and the dopant activation temperature. But as the first batch of Z-shape MRMs, its modulation efficiency is already close to that of other state-of-the-art MRMs. Further optimization of the Z-shape doping profile holds the potential for achieving higher modulation efficiency $V\pi\cdot L < 0.5 \text{ V}\cdot\text{cm}$.

Regarding the $V\pi\cdot L$ of $0.37 \text{ V}\cdot\text{cm}$ in Ref. [16], this modulation efficiency is more likely in the mixed region of carrier injection mode and carrier depletion mode. Figure 2(a) below shows the transmission spectrum of the MRM from 0.5V to -2.0V, and the wavelength shift decreases with higher reverse bias voltage. However, the reported $V\pi\cdot L \sim 0.37 \text{ V}\cdot\text{cm}$ is calculated from -0.5V to 0.5V. If using the depletion region wavelength shift value, the modulation efficiency $V\pi\cdot L > 0.53 \text{ V}\cdot\text{cm}$. To make it clearer, we have annotated this value in the comparison table. In addition, the previously reported bandwidth of 42 GHz was measured at -2 V. To be consistent with the measured bias voltage of reported $V\pi\cdot L$, we have changed the bandwidth to the measured value at 0 V, which is 35 GHz.

Fig. 1. (a) Top-view schematic of the Si RM (b) cross-section view for phase shifter

Fig. 2. (a) Measured transmission spectra at different bias voltage (b) Transmission penalty, extinction ratio and insertion loss

The U-shaped p-n junction design is quite ingenious, the longer depletion interface length enables a better modulation efficiency. As demonstrated in Ref. [30], the modulation efficiency $V\pi \cdot L$ is around $0.46 \text{ V} \cdot \text{cm}$, which is better than other junctions. However, this U-shaped junction is more suitable for low-speed MRMs. As shown in Fig. 1 below, the U-shaped junction has a sandwich structure, leading to a significantly larger junction capacitance area, and its junction length is nearly twice as long as that of the Z-shaped junction. The following increased RC time constant will limit the bandwidth of the MRM. One clever approach to address this issue is to decrease the doping concentration of the U-shaped junction, allowing the middle p-type layer to be depleted to substantially reduce the junction capacitance, as depicted in Fig. 1(b). However, this modification results in a much higher Q-factor, which, in turn, restricts the photon lifetime-limited bandwidth. This explains why in Ref. [30], only 13 Gb/s NRZ eye diagrams are presented. To conclude, this U-shaped junction idea is an effective way to improve the MRM performance at low data rates. Nevertheless, pushing it to higher data rates, such as 200 Gb/s, presents considerable challenges due to the inherent trade-offs between the junction design and bandwidth limitations.

Fig. 1. Cross-sections of the doping concentration profiles computed using Sentaurus TCAD of the U-shaped PN junction under bias voltages of (a) 0V and (b) -1V. The vertical profiles of the electron and hole densities at $x = 0 \mu\text{m}$ under bias voltages of (c) 0 V and (d) -1 V. The waveguide height is 150 nm, the slab height is 65 nm, and the rib width is 700 nm.

To provide better clarity and address your concerns, we have revised the comparison table, shown as: “

Ref.	Junction type	Modulation structure	Radius (μm)	Q	Bandwidth (GHz)	$V_{\pi} \cdot L$ (V·cm)	Operating λ (nm)	Data rate (Gb/s)	Energy cost (fJ/bit)
14	vertical	one-segment	4.8	6800	21	-	1550	1×44 (NRZ)	0.9
15	vertical	one-segment	5	6780	24	-	1310	1×40 (NRZ)	4
16	vertical	one-segment	5	4800	35	0.37 [†]	1310	1×100 (PAM4)	-
17	vertical	one-segment	8	5600	67 (-3dB IL)	0.8	1310	1×200 (PAM4)	-
18	lateral	one-segment	3	1050	67	0.94	1550	1×200 (PAM4)	-
19	lateral	one-segment	8	4500	52	0.825	1310	1×240 (PAM8)	-
20	lateral	two-segment	12	7500	18	-	1310	1×40 (PAM4)	-
21	lateral	two-segment	12	3600	40	1.0	1310	1×100 (PAM4)	9.9
22	L-shape	one-segment	10	5000	50	0.52	1310	1×128 (PAM4)	18
23	L-shape	one-segment	4	4000	54	0.53	1310	1×240 (PAM4)	-
30	U-shape	one-segment	32.5	26200	13.5	0.46	1310	1×13 (NRZ)	-
31	Z-shape	one-segment	9	7000	25 (-6dB IL) 42 (-3dB IL)	0.92	1310	1×100 (PAM4)	14
this work	Z-shape	two-segment	12	3700	48.6 (-6dB IL) 58.1 (-3dB IL)	0.6	1310	5×200* (PAM4)	6.3

[†] 0.37 V·cm is measured from 0.5 V to -0.5 V; $V_{\pi} \cdot L$ would increase to > 0.53 V·cm in the higher reverse voltage region.

* 200 Gb/s limited by the bandwidth of the experimental setup.

Table 1. Properties of depletion mode MRMs based on different types of Si p-n junctions.

”

The revised table adds an annotation on the outlier $V_{\pi} \cdot L$ in Ref. [16], changes the bandwidth from 42 GHz to 35 GHz in Ref. [16], and incorporates the properties of the U-shaped MRM in

Ref. [30]. And we have rewritten the **Discussion** section as “In this work, a Si MRM with two-segment Z-shape junctions has been demonstrated for use in the next-generation optical interconnects. The properties comparison of depletion mode MRMs with diverse Si p-n junctions, including vertical, lateral, L-shape, U-shape, and Z-shape junctions, are summarized in Table 1. To enhance modulation efficiency, longer depletion regions have been adopted for better mode overlap, such as the vertical, L-shape, and U-shape p-n junctions. The U-shape junction offers the best modulation efficiency $V\pi\cdot L$ of 0.46 V·cm. However, it also results in the longest junction capacitor, nearly twice as long as the L-shape and Z-shape junctions. The ensuing large RC time constant will hinder the high-speed operation. One potential solution to mitigate this RC issue is to lower the doping concentrations in the U-shape junction, allowing the middle p-type layer to be almost entirely depleted, thereby significantly reducing the junction capacitance³⁰. Unfortunately, the low doping concentrations result in a much higher Q factor, which in turn restricts the photon lifetime-limited bandwidth. Consequently, the U-shape design is more suitable for low-speed, high-efficiency modulation. Other than that, L-shape junctions exhibit the second-best modulation efficiency²². The simulated Z-shape junction is expected to offer ~ 11% better overlap compared to L-shape junctions, but the experimental results do not fully reflect this. This discrepancy can be attributed to the fact that the actual fabricated Z-shaped doping profile deviates from the ideal design. As the first batch of Z-shape MRMs, its modulation efficiency is already close to that of other state-of-the-art MRMs. Further optimization of the Z-shape doping profile holds the potential for achieving higher modulation efficiency $V\pi\cdot L < 0.5$ V·cm. Besides the enhanced overlap between the waveguide mode and the junction depletion region, our innovative design offers high bandwidth at the same time. The 4 distinct implantations create higher doping concentrations inside the slab region than the waveguide core region to achieve a reduced series resistance without leading to high free carrier absorption loss. With this unique degree of freedom, our Z-shape MRM exhibits a similarly high bandwidth even with larger microring radius. Meanwhile, the two-segment structure is employed to further reduce the capacitance of the Z-shape junctions. Therefore, this MRM combines a high RC-limited bandwidth with better modulation efficiency. In addition, the two-segment architecture simplifies the complex PAM4 driving signals into two NRZ signals. With the adoption of this optical DAC, the MRM can modulate higher-speed data with better linearity and lower CMOS driver energy consumption.

Thanks to the innovative design, our two-segment Z-shape MRM exhibits a high 3dB bandwidth of ~ 48.6 GHz at the wavelength with maximum modulation slope, a large modulation efficiency with $V\pi\cdot L$ of ~ 0.6 V·cm, and thus supports open 200 Gb/s PAM4 eye diagrams with only 1.6 V V_{pp} and 6.3 fJ/bit energy consumption. The demonstrated MRM achieves state-of-the-art performance for Si depletion mode modulators. Compared to the previous leading two-segment MRM with a lateral junction²¹, it enables a ~ 21 % increase in bandwidth, a remarkable ~ 67 % enhancement in modulation efficiency, a doubling of the data rate, and a ~ 36% reduction in energy cost per bit. With this high performance, the 5-channel DWDM Si MRM array based on two-segment Z-shape junctions has been realized, which supports 1 Tb/s (5×200 Gb/s PAM4) data rates. The inherent offset resonances of the microrings allow the DWDM MRM array to act as an effective multiplexer (MUX), achieving an impressively low crosstalk of < -33 dB directly, with the potential for further improvement to ~ -57 dB with control circuits. The DWDM structure provides scalability for ever-increasing data rates. By reducing the MRM radius to 5 μm , the FSR of the MRM can be extended to ~ 13.7 nm with a bend loss of only about 0.06 dB. With the same channel spacing, 12 channels can be supported, easily scaling up the full data rate

to 2.4 THz. To encapsulate, this work showcases a total of 1 Tb/s DWDM modulation integrated on a Si chip with a small V_{pp} of 1.6 V, paving the way to satisfy the ever-increasing data demands for intr- and inter-chip interconnects.”

[30] Yong, Z. et al. U-shaped pn junctions for efficient silicon mach-zehnder and microring modulators in the o-band. *Opt. express* **25**, 8425–8439 (2017).

2. The electro-optic (EO) bandwidth and the data rate achieved with the MRM do not rank as the best when compared to some of the previously published results (as shown in table 1). A strong justification is needed for this paper to be published in Nature Communications.

Response: Thank you for the valuable suggestion. Similar to L-shape and U-shape junctions, the Z-shape junction sacrifices the junction capacitance to achieve a better carrier-light interaction. To address the worse RC bandwidth, we used four doping regions to reduce the series resistance and a two-segment structure to reduce the junction capacitance. As a result, this Z-shape two-segment MRM can achieve high bandwidth and high modulation efficiency simultaneously. As shown in table 1, there are several MRMs enable higher EO bandwidth than us, but most of these devices have much worse modulation efficiency than ours. It is also worth noting that some references reported bandwidths at large wavelength detuning to take the advantage of the optical peaking effect. In order to clarify this, we have modified Table 1 to include bandwidths at insertion losses of -6 dB and -3 dB.

The only MRM that shows better performance than ours is Intel’s L-shape device [23]. However, this higher EO bandwidth is due to the fact that it has a much smaller radius than ours, only 4 μm . Therefore, it can effectively suppress the junction capacitance. The reduced junction capacitance can boost the RC time bandwidth considering the entire circuit. If we shrink the radius of the MRM from 12 μm to 4 μm , the simulated RC responses of the LSB and MSB are illustrated in Figure below. Due to the smaller junction capacitance, the RC time-limited bandwidths of the LSB and MSB extend to ~ 94.3 GHz and 86.3 GHz, respectively; an improvement of about 19% for the LSB and 34% for the MSB. Using the simplified equation, $f_{est} = f_{RC}f_{ph}/\sqrt{f_{RC}^2 + f_{ph}^2}$, the estimated EO bandwidth of both LSB and MSB will be > 50 GHz. The Z-shape MRM has already achieved state-of-the-art speed performance in this large device radius.

Figure S3. Simulated RC response of the (a) LSB and (b) MSB of the two-segment MRM with a 4 μm radius.

The data rate of our MRM is mainly limited by the eye diagram test setup. The 3dB bandwidth of the electrical power amplifiers, bias-tees, optical receiver, and electrical cables are 60 GHz, 50 GHz, 65 GHz, and 65 GHz respectively. The combination of these components results in a loss of ~ 10 dB in 50 GHz. In fact, we first measured the eye diagram using 50 GHz power amplifiers, and the highest data rate we could achieve was about 160 Gb/s. Therefore, we purchased 60 GHz power amplifiers to achieve 200 Gb/s. Besides that, according to the M8194A data sheet, the 120 GSa/s arbitrary waveform generator (AWG) can generate frequency content up to 50 GHz. The 100 Gb/s NRZ signals are the upper limit of the AWG, our MRMs have pushed the experimental setup to its limitation. To further improve the eye diagram, a higher speed experimental setup is required.

To address reviewer's concerns, we have modified the Table 1 as “

Ref.	Junction type	Modulation structure	Radius (μm)	Q	Bandwidth (GHz)	$V_{\pi} \cdot L$ (V \cdot cm)	Operating λ (nm)	Data rate (Gb/s)	Energy cost (fJ/bit)
14	vertical	one-segment	4.8	6800	21	-	1550	1 \times 44 (NRZ)	0.9
15	vertical	one-segment	5	6780	24	-	1310	1 \times 40 (NRZ)	4
16	vertical	one-segment	5	4800	35	0.37 [†]	1310	1 \times 100 (PAM4)	-
17	vertical	one-segment	8	5600	67 (-3dB IL)	0.8	1310	1 \times 200 (PAM4)	-
18	lateral	one-segment	3	1050	67	0.94	1550	1 \times 200 (PAM4)	-
19	lateral	one-segment	8	4500	52	0.825	1310	1 \times 240 (PAM8)	-
20	lateral	two-segment	12	7500	18	-	1310	1 \times 40 (PAM4)	-
21	lateral	two-segment	12	3600	40	1.0	1310	1 \times 100 (PAM4)	9.9
22	L-shape	one-segment	10	5000	50	0.52	1310	1 \times 128 (PAM4)	18
23	L-shape	one-segment	4	4000	54	0.53	1310	1 \times 240 (PAM4)	-
30	U-shape	one-segment	32.5	26200	13.5	0.46	1310	1 \times 13 (NRZ)	-
31	Z-shape	one-segment	9	7000	25 (-6dB IL) 42 (-3dB IL)	0.92	1310	1 \times 100 (PAM4)	14
this work	Z-shape	two-segment	12	3700	48.6 (-6dB IL) 58.1 (-3dB IL)	0.6	1310	5 \times 200* (PAM4)	6.3

[†] 0.37 V \cdot cm is measured from 0.5 V to -0.5 V; $V_{\pi} \cdot L$ would increase to > 0.53 V \cdot cm in the higher reverse voltage region.

* 200 Gb/s limited by the bandwidth of the experimental setup.

Table 1. Properties of depletion mode MRMs based on different types of Si p-n junctions.

”;

modified “The maximum modulation slope of the MRM occurs at the wavelength detuning of $0.29 \times \text{FWHM}$. Given the resonance width FWHM of 0.35 nm for this MRM, the wavelength detuning we choose for the following measurements is set at $\Delta\lambda$ of 0.1 nm (~ 17.5 GHz), which is located at the -6 dB insertion loss (IL) point on the transmission spectrum. Figure 3 (c) and (f) display the measured EO responses ($|S_{21}|^2$) for the LSB and MSB segments, respectively. The LSB junction has a 3 dB EO bandwidth of ~ 48.9 GHz, and the EO bandwidth is ~ 48.3 GHz for the MSB junction. These two EO bandwidth values align with the estimated bandwidth from the RC-limited bandwidth and photon lifetime-limited bandwidth, $f_{est} = f_{RC}f_{ph}/\sqrt{f_{RC}^2 + f_{ph}^2}$. Using the optical peaking effect, the EO bandwidth of the MRM can be further increased. For instance, this MRM exhibits a 3 dB bandwidth of ~ 58 GHz at -3 dB IL, as shown in Supplementary Information II.” in the **Single MRM: RF characterizations** section on Page 5;

added “It is worth noting that 200 Gb/s eyes are the upper limit of the experimental setup, higher bandwidth setup can further improve the data rate.” to the **DWDM MRMs** section on Page 6;

added the **Electro-optical response** section in **Supplementary Information** “A Z-shape Si p-n junction has been adopted in the MRM to enhance the overlap between the carrier variance and optical mode. However, the increased depletion interface of the Z-shape junction leads to a larger junction capacitance, which ultimately limits the overall bandwidth of the MRM. To address this trade-off, four dopant implants were applied to the microring waveguide. This design provides an additional degree of freedom to independently adjust the doping concentrations in the slab region and the waveguide core region. Therefore, high modulation efficiency, reasonable free-carrier absorption, and small series resistance can be achieved simultaneously. The small series resistance effectively alleviates the trade-off between modulation efficiency and bandwidth. To attain the next-generation data rate of 200 Gb/s per lane, just reducing the series resistance is insufficient to meet the required electro-optical (EO) bandwidth. Fortunately, the optical digital-to-analog converter (DAC), the two-segment structure, can further improve the RC bandwidth. Figure 3 (a) and (d) show the measured S11 charts for the least significant bit (LSB) and the most significant bit (MSB) segments at -3 V. The equivalent circuits for both segments can be derived through the S11, shown as insets in Fig. 3 (b) and (e). Although the product of the junction capacitance and series resistance, $C_j \times R_s$, is almost independent of the junction length, the smaller capacitance still provides higher RC bandwidth considering the entire equivalent circuits. The LSB and MSB junctions have RC bandwidths of ~ 79.1 GHz and 64.5 GHz, respectively. As a comparison, the equivalent one-segment MRM would yield a much smaller RC bandwidth of ~ 53.6 GHz.

The measured EO bandwidths of both segments, as depicted in Fig. S2, are close to each other. This is because the RC bandwidths of both segments are higher than the photon lifetime-limited bandwidth $f_{ph} \sim 62$ GHz. The f_{ph} dominates the EO bandwidth, therefore the overall bandwidth does not vary much. Differently, if the RC bandwidth is less than f_{ph} , it will exhibit a larger influence. For instance, with the one-segment MRM featuring a 53.6 GHz RC bandwidth, the overall EO bandwidth will reduce to ~ 41 GHz. As a result, the two-segment design improves the RC bandwidth by $> 20\%$ and the consequent EO bandwidth by $> 17\%$.

Figure S2. Measured EO response ($|S_{21}|^2$) of the two-segment MRM at insertion loss of (a) -6 dB and (b) -3 dB.

The EO response of the MRM is also affected by wavelength detuning. The MRM can achieve an extended 3 dB bandwidth by tuning the wavelength away from the resonance. Figure S2(a) is the measured EO response of the LSB and MSB segments at the wavelength corresponding to the maximum modulation slope, where the wavelength detuning $\Delta\lambda$ is ~ 0.1 nm. This wavelength is located at the -6 dB insertion loss (IL) point on the transmission spectrum of the MRM. The LSB and MSB have 3 dB bandwidths of ~ 48.9 GHz and 48.3 GHz, respectively. By further detuning the wavelength towards the IL of -3 dB, a distinct optical peaking effect can be observed, as shown in Fig. S2(b). The measured results are limited to 50 GHz due to the bandwidth constraint of the vector network analyzer. The fitted curves indicate that the LSB exhibits an enhanced 3 dB bandwidth of ~ 59.1 GHz and the MSB attains a 3 dB bandwidth of ~ 57.1 GHz.

Thanks to the novel design, this MRM achieves state-of-the-art EO bandwidth with a relatively large radius of 12 μm . By further reducing the MRM radius, the smaller junction capacitance can enhance the EO bandwidth considering the entire equivalent circuit. If the radius reduces to 4 μm , the simulated RC responses of the equivalent circuits of the LSB and MSB are illustrated in Fig. S3 (a) and (b), respectively. As expected, the RC time-limited bandwidth of the LSB extends from ~ 79.1 GHz to ~ 94.3 GHz, and the MSB's RC bandwidth improves from ~ 64.5 GHz to ~ 86.3 GHz. Using the simplified EO bandwidth equation, $f_{est} = f_{RC}f_{ph}/\sqrt{f_{RC}^2 + f_{ph}^2}$, the estimated EO bandwidth of both LSB and MSB will be > 50 GHz without detuning peaking effect, which is comparable to Intel's 4 μm -radius MRM.”;

and modified the **Eye diagrams measurements** section “The eye diagrams of the MRMs have been performed with a 120 GSa/s arbitrary waveform generator (AWG) M8194A, which can generate frequency content up to 50 GHz. It provides two PRBS9 NRZ signals for the LSB and MSB. A signal generator was used to generate a 2.5 GHz clock signal for the AWG clock reference, the digital communication analyzer (DCA) trigger reference, and the DCA precision reference. The highest swing voltage V_{pp} of the AWG is 0.8 V, therefore two identical 60 GHz electrical power amplifiers were added to amplify the swing voltage. After that, two identical 50 GHz bias-tees were connected to combine the RF signals and DC bias voltage. The measured V_{pp}

after the bias-tee at 100 Gb/s NRZ, i.e. the LSB and MSB driving voltage, is 1.6 V. A commercial tunable CW laser, Santec TSL 510, was used as the input optical source, and the MRM output light was amplified by a Praseodymium-doped fiber amplifier (PDFA) to compensate for the link losses. A 65 GHz optical module N1030A was used to receive the modulated signals on the DCA. The measurement setup is illustrated in Supplementary Information VI. The signal distortion caused by the power amplifier, bias-tee, and RF cables was calibrated by the AWG internal calibration tool. In order to eliminate the amplified spontaneous emission (ASE) noise from the PDFA, the received patterns were averaged 64 times. A more than half-baud rate 4th-order Bessel filter and a 21-tap FFE were used at the receiver. The TDECQ values of the 200 Gb/s PAM4 eye diagrams were measured at an SD-FEC threshold of SER at 1E-2. The 200 Gb/s PAM4 is the upper limit of the current eye diagram experimental setup, a higher data rate is achievable with higher bandwidth AWG and RF components.”

3. The claimed data rate of 1 Tbps is a projected capability of the MRM array rather than an experimentally demonstrated data transmission with all 5 MRMs in operation. Additionally, the crosstalk measurements do not reflect real operational conditions, where each MRM would be carrying 200 Gbps of data.

Response: Thank you very much for your constructive comment. Currently, the MRM array is measured using the setup shown in Fig. S8. The AWG has only two high-speed channels, thus an external signal generator is required to synchronize AWG and DCA. Additionally, there are only two high-speed electrical amplifiers in our lab and the customized probe can only probe one two-segment MRM at a time. Due to the above limitations, the data rate of 1 Tb/s is measured channel by channel. The only thing we changed during the DWDM measurement was the input laser wavelength and probe position. Although the MRM array was measured channel by channel, the consistent performance of 5 channels indicates that the photonic chip has the capability to support 1 Tb/s. By implementing a customized CMOS driving circuit, the 5 channels can operate simultaneously.

However, you are correct that it would be desirable to see more than one channel operation to further support the conclusions of 1 Tb/s data rate and low channel crosstalk. To address this concern, we came up with a way to measure two channels simultaneously using our limited setup. The new measured setup is shown in Fig. S6(a), we shifted half a channel to probe the channel 2 MSB and channel 3 LSB at the same time. Therefore, even with the limited two high-speed driving signals, we can still measure two channels of the MRM array simultaneously. 100 Gb/s NRZ drive signals were applied to channel 2 MSB and channel 3 LSB to demonstrate the effects of multi-channel operation at 100 Gbaud/s data rates. The 100 Gb/s NRZ eye diagrams were measured for channel 2 MSB (Fig. S6(b) and (c)) and channel 3 LSB (Fig. S6(d) and (e)), respectively. Toggling the driving signal off and on of its neighboring channel does not impact the signal-to-noise ratios (SNRs) of the measured eyes. This result serves as compelling proof that the MRM array exhibits negligible crosstalk under 100 Gbaud/s modulation conditions. Consequently, this DWDM MRM chip is entirely feasible to support a total data rate of 1 Tb/s.

To address this comment, we have added the section **Experimental Setup for eye measurement** in **Supplementary Information** as “The experimental setup for eye diagram measurement is

shown in Fig. S8. A dual-channel 120 GSa/s arbitrary waveform generator (AWG) M8194A was used to provide NRZ signals for the LSB and MSB. It can generate frequency content up to 50 GHz. An external signal generator was used to provide 2.5 GHz square wave signals to synchronize the AWG and the DCA. The two NRZ driving signals were amplified by two identical 60 GHz electrical power amplifiers (SHF S804 B), passed through two identical 50 GHz bias-tees (11612B), and provided 1.6 V swing voltages on both LSB and MSB. The driving signals were calibrated using AWG internal calibration function to compensate for the response of the amplifiers, bias-tees, and RF cables. By delaying integer bits between the two driving NRZ signals, the two-segment MRM can generate optical PAM4 signals. A customized probe was used to probe one MRM at a time. The five 200 Gb/s PAM4 eye diagrams were measured channel by channel. Due to the 50 GHz bandwidth of the AWG and bias-tees, 200 Gb/s PAM4 is the upper limit of this experimental setup. By increasing the bandwidth of the setup, the quality of the eye diagrams and the data rate can be further improved.

Figure S8. Experimental setup for eye diagram measurement.”

and added the section **Linearity and crosstalk** in **Supplementary Information** as “The segment length ratio of ~ 2:1 makes the MRM an optical DAC to simplify the pulse amplitude modulation with four levels (PAM4) driving signal to two non-return-to-zero (NRZ) driving signals. The equally spaced four levels are critical to realize PAM4 modulation. Figure S4(a) illustrates the measured optical transmission spectrum of the MRM at four bias levels: 1) LSB = 0 V, MSB = 0 V; 2) LSB = -4 V, MSB = 0 V; 3) LSB = 0 V, MSB = -4 V; and 4) LSB = -4 V, MSB = -4V. The resonant wavelength of the MRM red shifts with bias voltage levels. To quantify the linearity of the optical DAC, the measured spectrum on a linear scale is also plotted as shown in Fig. S4(b). At a fixed laser wavelength indicated by the black dash line, the optical

power differences of the four bias levels are about 0.16, 0.18, and 0.16, respectively, with a maximum spacing deviation of $\sim 8\%$. This two-segment optical DAC has good linearity and enables nearly equally spaced PAM4 modulation.

Figure S4. Measured transmission spectrum of the two-segment MRM at four bias levels on (a) dB scale and (b) linear scale.

In addition to measuring optical transmission intervals at DC bias voltages, the RF responses of LSB, MSB, and their combination were also conducted to demonstrate the linearity in large-signal RF cases. Figure S5 displays the captured waveforms on the digital communication analyzer (DCA) at 50 Gbaud/s data rate, where the pink waveform indicates driving only LSB, the yellow waveform corresponds to driving only MSB, and the blue curve represents the simultaneous driving of both LSB and MSB. If there is no time offset between the driving signals of LSB and MSB, all three waveforms should exhibit similar shapes. As shown in Fig. S5(a), three curves share similar waveforms. Moreover, the optical modulation amplitude (OMA) ratio of the three waveforms is approximately 1:2:3, underscoring great linearity in the RF domain. It is worth noting that the driving signals and swing voltages are identical for both LSB and MSB. An advantage of the two-segment design is the ability to independently tune the two driving signals, allowing for the possibility of achieving even better linearity by making slight adjustments to the amplitudes of the driving signals. On the other hand, by introducing an integer-bits offset between the driving signals of LSB and MSB, the combined response should exhibit four distinct levels of modulation. Figure S5(b) presents the captured waveforms with a 20 ps offset, i.e., 1 bit, and an example six-bit sequence is depicted by the red dash lines. The LSB waveform shows a sequence of [0, 1, 1, 0, 1, 0], whereas the MSB waveform is 1 bit later with a sequence of [0, 0, 1, 1, 0, 1]. The combined waveform of both LSB and MSB displays a sequence of [0, 1, 3, 2, 1, 2], aligning with the requirements of the optical DAC. This result not only demonstrates the good linearity of the large-signal RF response but also indicates the absence of noticeable electrical crosstalk.

Figure S5. Measured waveforms of LSB, MSB, and LSB + MSB at 50 Gbaud/s data rate: (a) No offset between LSB and MSB drive signals. (b) 20 ps offset (i.e., 1 bit) between LSB and MSB drive signals.

Due to the limited channel number in the current setup, we can only provide two high-speed driving signals to this MRM array. In order to closely simulate the real product operating condition, which involves driving multiple channels simultaneously using custom complementary metal-oxide-semiconductor (CMOS) driver circuits, we shifted half a channel to evaluate optical crosstalk. The micrograph of the measured MRM array is shown in Fig. S6(a), the customized probe simultaneously probes channel 2 MSB and channel 3 LSB, rather than probing LSB and MSB of one channel. Figure S6(b) and (c) show the measured 100 Gb/s NRZ eye diagrams of the channel 2 MSB. The input laser wavelength is aligned with channel 2, channel 2 MSB driving signal remains on, while the channel 3 LSB driving signal is off (left) and on (right). The signal-to-noise ratios (SNRs) of the two 100 Gb/s NRZ eyes are very close, around 3.1. Likewise, the 100 Gb/s NRZ eye diagrams of the channel 3 LSB are presented in Fig. S6 (d) and (e). Toggling the driving signal off and on of its neighboring channel 2 does not impact the eye quality, both eye diagrams have an SNR of ~ 2.9 . Hence, this measurement serves as compelling proof that the dense wavelength division multiplexing (DWDM) MRM array exhibits negligible optical crosstalk under 100 Gbaud/s modulation conditions. Consequently, it is entirely feasible to support a total data rate of 1 Tb/s with this DWDM MRM array.

Figure S6. (a) Micrograph of the probed MRM array for large signal crosstalk measurements. Measured 100 Gb/s NRZ eye diagrams: input wavelength is λ_{C2} , C2 driving signal is on, C3 driving signal is (b) off and (c) on; input wavelength is λ_{C3} , C3 driving signal is on, C2 driving signal is (d) off and (e) on.”

4. The authors should review and consider these points to improve the quality and credibility of the paper. This will also help in justifying its publication in Nature Communications over other technical journals.

Response: We would like to thank you once again for your valuable comments and suggestions! These suggestions have helped us to improve the quality of the paper considerably, making the work more solid and persuasive.

REVIEWER COMMENTS

Reviewer #1 (Remarks to the Author):

The authors have addressed the comments very well. I suggest to accept it with the current version.

Reviewer #2 (Remarks to the Author):

The author has addressed all the concerns sufficiently, and I support the manuscript to be published in Nature Communications.

Reviewer #3 (Remarks to the Author):

The authors have thoughtfully addressed and clarified the queries and concerns raised by the reviewers, providing insights into the limitations imposed by both fabrication and measurement equipment on the modulator's performance. It becomes apparent that the desired performance of the MRM-based DWDM transmitter has not been fully substantiated.

The designs of micro-ring modulators are intricate, requiring careful consideration of numerous trade-offs. Designing for manufacturability is a pivotal element in modulator design. While numerous p-n junction designs have been proposed in the past, few have been successfully manufactured to meet the simulated design targets. A persuasive experimental demonstration is crucial to establish the design's superiority over existing state-of-the-art technologies.

Considering that the performance demonstrations have focused solely on individual micro-ring modulators, with no simultaneous operation of all MRMs, the use of "1 Tb/s DWDM Modulator" in the title appears inappropriate and unjustified. I recommend that the authors omit "1 Tb/s" from the title and instead discuss the potential in the concluding discussion section."

Reviewer #3

The authors have thoughtfully addressed and clarified the queries and concerns raised by the reviewers, providing insights into the limitations imposed by both fabrication and measurement equipment on the modulator's performance. It becomes apparent that the desired performance of the MRM-based DWDM transmitter has not been fully substantiated.

The designs of micro-ring modulators are intricate, requiring careful consideration of numerous trade-offs. Designing for manufacturability is a pivotal element in modulator design. While numerous p-n junction designs have been proposed in the past, few have been successfully manufactured to meet the simulated design targets. A persuasive experimental demonstration is crucial to establish the design's superiority over existing state-of-the-art technologies.

Considering that the performance demonstrations have focused solely on individual micro-ring modulators, with no simultaneous operation of all MRMs, the use of "1 Tb/s DWDM Modulator" in the title appears inappropriate and unjustified. I recommend that the authors omit "1 Tb/s" from the title and instead discuss the potential in the concluding discussion section.

Response: Thank you for your thorough suggestion. We agree with your concern that the 5 channels of the DWDM modulators are not simultaneously operating so the "1Tb/s" might be inappropriate. Considering that the main results of this work are the 5×200 Gb/s data rate and we have demonstrated that the channel crosstalk is small for the simultaneous operation, the capability of realizing 1Tb/s data rate is feasible.

To provide a more accurate representation of the experimental results, the title of the manuscript is revised. It omits "1 Tbps" and instead uses "5×200 Gbps", which can precisely reflect the measured 200 Gb/s data rate on all 5 channels of the DWDM microring modulators. The revised title now is "A 5×200 Gbps DWDM Microring Modulator Silicon Chip Empowered by Two-Segment Z-Shape Junctions".